# A Deep Top-Down Approach to Hierarchically Coherent Probabilistic Forecasting

## Abstract

Probabilistic, hierarchically coherent forecasting is a key problem in many practical forecasting applications – the goal is to obtain coherent probabilistic predictions for a large number of time series arranged in a pre-specified tree hierarchy. In this paper, we present a probabilistic top-down approach to hierarchical forecasting that uses a novel attention-based LSTM model to learn the distribution of the proportions according to which each parent prediction is split among its children nodes at any point in time. These probabilistic proportions are then coupled with an independent univariate probabilistic forecasting model for the root time series. The resulting forecasts are naturally coherent, and provide probabilistic predictions over all time series in the hierarchy. We experiment on several public datasets and demonstrate significant improvements up to 27% on most datasets compared to state-of-the-art probabilistic hierarchical models. Finally, we also provide theoretical justification for the superiority of our top-down approach compared to traditional bottom-up modeling.

## 1 Introduction

A central problem in multivariate forecasting is the need to forecast a large group of time series arranged in a natural hierarchical structure, such that time series at higher levels of the hierarchy are aggregates of time series at lower levels. For example, hierarchical time series are common in retail forecasting applications (Fildes et al., 2019), where the time series may capture retail sales of a company at different granularities such as item-level sales, category-level sales, and department-level sales. In electricity demand forecasting (Van Erven & Cugliari, 2015), the time series may correspond to electricity consumption at different granularities, starting with individual households, which could be progressively grouped into city-level, and then state-level consumption time-series. The hierarchical structure among the time series is usually represented as a tree, with leaf-level nodes corresponding to time series at the finest granularity, while higher-level nodes represent coarser-granularities and are obtained by aggregating the values from its children nodes.

Since businesses usually require forecasts at various different granularities, the goal is to obtain accurate forecasts for time series at every level of the hierarchy. Furthermore, to ensure decision-making at different hierarchical levels are aligned, it is essential to generate predictions that are *coherent* (Hyndman et al., 2011) with respect to the hierarchy, that is, the forecasts of a parent time-series should be equal to the sum of forecasts of its children time-series [1]. Finally, to facilitate better decision making, there is an increasing shift towards probabilistic forecasting (Berrocal et al., 2010; Gneiting & Katzfuss, 2014); that is, the forecasting model should quantify the uncertainty in the output and produce probabilistic predictions.

In this paper, we address the problem of obtaining coherent probabilistic forecasts for large-scale hierarchical time series. While there has been a plethora of work on multivariate forecasting, there is significantly limited research on multivariate forecasting for hierarchical time series that satisfy the requirements of both hierarchical coherence **and** probabilistic predictions.

There are numerous recent works on deep neural network-based multivariate forecasting (Salinas et al., 2020; Oreshkin et al., 2019; Rangapuram et al., 2018; Benidis et al., 2020; Sen et al., 2019; Olivares et al., 2022a),

---

[1]Note that this is a non-trivial constraint. For example, generating independent predictions for each time series in the hierarchy using a standard multivariate forecasting model does not guarantee coherent predictions.

including probabilistic multivariate forecasting (Salinas et al., 2019; Rasul et al., 2021) and even graph neural network(GNN)-based models for forecasting on time series with graph-structure correlations (Bai et al., 2020; Cao et al., 2020; Yu et al., 2017; Li et al., 2017). However, none of these works ensure coherent predictions for hierarchical time series.

On the other hand, several papers specifically address hierarchically-coherent forecasting (Hyndman et al., 2016; Taieb et al., 2017; Van Erven & Cugliari, 2015; Hyndman et al., 2016; Ben Taieb & Koo, 2019; Wickramasuriya et al., 2015; 2020; Mancuso et al., 2021; Abolghasemi et al., 2019). Most of them are based on the idea of reconciliation. This involves a two-stage process where the first stage generates independent (possibly incoherent) univariate base forecasts, and is followed by a second "reconciliation" stage that adjusts these forecasts using the hierarchy structure, to finally obtain coherent predictions. These approaches are usually disadvantaged in terms of using the hierarchical constraints only as a post-processing step, and not during generation of the base forecasts. Furthermore, none of these approaches can directly handle probabilistic forecasts.

To the best of our knowledge, the only prior work on hierarchically coherent probabilistic forecasting are the proposals in Rangapuram et al. (2021); Olivares et al. (2022b) and Taieb et al. (2017). While Taieb et al. (2017) is a two-stage reconciliation-based model for coherent probabilistic hierarchical forecasting, Rangapuram et al. (2021); Olivares et al. (2022b) directly incorporates a differentiable reconciliation step as part of a deep neural network-based training process (by projecting into a linear subspace that satisfies the hierarchical constraints).

In this paper, we present an alternate approach to deep probabilistic forecasting for hierarchical time series, motivated by a classical method that has not received much recent attention: top-down forecasting. The basic idea is to first model the top-level forecast in the hierarchy tree, and then model the ratios or proportions according to how the top level forecasts should be distributed among the children time-series in the hierarchy. The resulting predictions are naturally coherent. Early top-down approaches were non-probabilistic, and were rather simplistic in terms of modeling the proportions; for example, by obtaining the proportions from historical averages (Gross & Sohl, 1990), or deriving them from independently generated (incoherent) forecasts of each time-series from another model (Athanasopoulos et al., 2009). In this paper, we showcase how modeling these proportions more effectively as part of a deep, probabilistic, top-down approach can outperform state-of-the-art probabilistic, hierarchically-coherent models.

Crucially, our proposed model (and indeed all top-down approaches for forecasting) relies on the intuition that the top level time series in a hierarchy is usually much less noisy and less sparse, and hence much easier to predict. Furthermore, it might be easier to predict proportions (that are akin to scale-free normalized time-series) at the lower level nodes than the actual time series themselves.

Our approach to top-down probabilistic forecasting involves modeling the proportions with a single end-to-end deep learning model that jointly forecasts the proportions along which each parent time series is disaggregated among its children. We use a Dirichlet distribution (Olkin & Rubin, 1964) to model the distribution of proportions for each parent-children family in the hierarchy. The parameters of the Dirichlet distribution for each family is obtained from a LSTM (Hochreiter & Schmidhuber, 1997) with multi-head self-attention (Vaswani et al., 2017), that is jointly learnt from all the time-series in the hierarchy.

Our model can be coupled with any univariate probabilistic forecast for the root time series, to immediately obtain coherent probabilistic forecasts for the entire hierarchy tree. The (univariate) root time series can be modeled independently - this flexibility is an added benefit of our method as any advancement in probabilistic, univariate time-series forecasting can be incorporated seamlessly in our framework. In particular, our experimental results use Prophet (Taylor & Letham, 2018), a simple, off-the-shelf (non-deep-learning) univariate package.

We validate our model against state-of-the art probabilistic hierarchical forecasting baselines on six public datasets, and demonstrate significant gains using our approach, outperforming all baselines on most datasets. In particular, we observe a relative improvement of as much as 27% over all baselines in terms of CRPS scores on one of the largest publicly available hierarchical forecasting datasets.

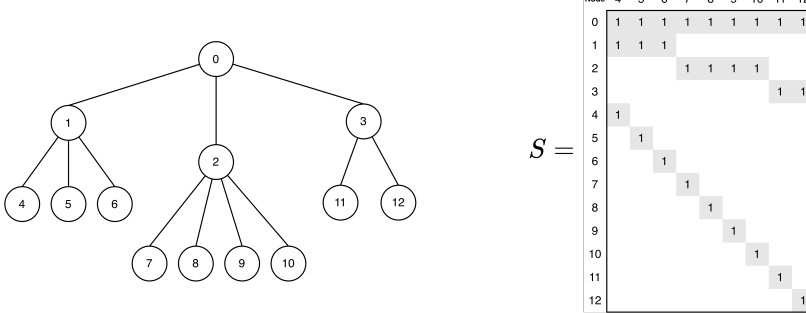

Figure 1: An example of a hierarchy and its corresponding $\boldsymbol{S}$ matrix. The rows and columns of the matrix are indexed by the corresponding nodes for easier interpretation. The empty cells of the matrix are zeros, and hence omitted from the figure.

Additionally, we theoretically analyze the advantage of the top-down approach (over a bottom-up approach) in a simplified regression setting for hierarchical prediction, and thereby provide theoretical justification for our top-down model. Specifically, we prove that for a 2-level hierarchy of $d$-dimensional linear regression with a single root node and $K$ children nodes, the excess risk of the bottom-up approach is $\min(K, d)$ time bigger than the one of the top-down approach in the worst case. This validates our intuition that it is easier to predict proportions than the actual values.

## 2 Background and Related Work

Hierarchical forecasting is a multivariate forecasting problem, where we are given a set of $N$ univariate time-series (each having $T$ time points) that satisfy linear aggregation constraints specified by a predefined hierarchy. More specifically, the data can be represented by a matrix $\boldsymbol{Y} \in \mathbb{R}^{T \times N}$, where $\boldsymbol{y}^{(i)}$ denotes the $T$ values of the $i$-th time series, $\boldsymbol{y}_t$ denotes the values of all $N$ time series at time $t$, and $y_{t,i}$ the value of the $i$-th time series at time $t$. We will assume that $y_{i,t} \geq 0$, which is usually the case in all retail demand forecasting datasets (and is indeed the case in all public hierarchical forecasting benchmarks (Wickramasuriya et al., 2015)). We compactly denote the $H$-step history of $\boldsymbol{Y}$ by $\boldsymbol{Y}_{\mathcal{H}} = [\boldsymbol{y}_{t-H}, \cdots, \boldsymbol{y}_{t-1}]^\top \in \mathbb{R}^{H \times N}$ and the $H$-step history of $\boldsymbol{y}^{(i)}$ by $\boldsymbol{y}_{\mathcal{H}}^{(i)} = [y_{t-H}^{(i)}, \cdots, y_{t-1}^{(i)}] \in \mathbb{R}^H$. Similarly we can define the $F$-step future as $\boldsymbol{Y}_{\mathcal{F}} = [\boldsymbol{y}_t, \cdots, \boldsymbol{y}_{t+F-1}]^\top \in \mathbb{R}^{F \times N}$. We use the $\hat{\cdot}$ notation to denote predicted values, and the $\cdot^\top$ notation to denote the transpose. We denote the matrix of external covariates like holidays etc by $\boldsymbol{X} \in \mathbb{R}^{T \times D}$, where the $t$-th row denotes the $D$-dimensional feature vector at the $t$-th time step. For simplicity, we assume that the features are shared across all time series[2]. We define $\boldsymbol{X}_{\mathcal{H}}$ and $\boldsymbol{X}_{\mathcal{F}}$ as above. The $\hat{\cdot}$ notation will be used to denote predicted values, for instance $\hat{\boldsymbol{Y}}_{\mathcal{F}}$ denotes the prediction in the future.

**Hierarchy:** The $N$ time series are arranged in a tree hierarchy, with $m$ leaf time-series, and $k = N - m$ non-leaf (or aggregated) time-series that can be expressed as the sum of its children time-series, or alternatively, the sum of the leaf time series in its sub-tree. Let $\boldsymbol{b}_t \in \mathbb{R}^m$ be the values of the $m$ leaf time series at time $t$, and $\boldsymbol{r}_t \in \mathbb{R}^k$ be the values of the $k$ aggregated time series at time $t$. The hierarchy is encoded as an aggregation matrix $\boldsymbol{R} \in \{0, 1\}^{k \times m}$, where an entry $R_{ij}$ is equal to 1 if the $i$-th aggregated time series is an ancestor of the $j$-th leaf time series in the hierarchy tree, and 0 otherwise. We therefore have the aggregation constraints $\boldsymbol{r}_t = \boldsymbol{R}\boldsymbol{b}_t$ or $\boldsymbol{y}_t = [\boldsymbol{r}_t^\top \ \boldsymbol{b}_t^\top]^\top = \boldsymbol{S}\boldsymbol{b}_t$ where $\boldsymbol{S}^T = [\boldsymbol{R}^\top | \boldsymbol{I}_m]$. Here, $\boldsymbol{I}_m$ is the $m \times m$ identity matrix. Such a hierarchical structure is ubiquitous in multivariate time series from many domains such as retail, traffic, etc, as discussed earlier. We provide an example tree with its $\boldsymbol{S}$ matrix in Figure 1. Note that we can extend this equation to the matrix $\boldsymbol{Y} \in \mathbb{R}^{T \times N}$. Let $\boldsymbol{B} := [\boldsymbol{b}_1; \cdots; \boldsymbol{b}_m]^\top$ be the corresponding leaf time-series values arranged in a $T \times m$ matrix. Then coherence property of $\boldsymbol{Y}$ implies that $\boldsymbol{Y}^\top = \boldsymbol{S}\boldsymbol{B}^\top$. Note that we will use $\boldsymbol{B}_{\mathcal{F}}$ to denote the leaf-time series matrix corresponding to the future time-series in $\boldsymbol{Y}_{\mathcal{F}}$.

---

[2]Note that our modeling can handle both shared and time-series specific covariates in practice.

**Coherency:** Clearly, an important property of hierarchical forecasting is that the forecasts also satisfy the hierarchical constraints $\hat{\boldsymbol{Y}}_{\mathcal{F}}^{\top} = \boldsymbol{S}\hat{\boldsymbol{B}}_{\mathcal{F}}^{\top}$. This is known as the *coherence* property which has been used in several prior works (Hyndman & Athanasopoulos, 2018; Taieb et al., 2017). Imposing the coherence property makes sense since the ground truth data $\boldsymbol{Y}$ is coherent by construction. Coherence is also critical for consistent decision making at different granularities of the hierarchy.

Our *objective* is to accurately predict the distribution of the future values $\hat{\boldsymbol{Y}}_{\mathcal{F}} \sim \hat{f}(\boldsymbol{Y}_{\mathcal{F}})$ conditioned on the history such that any sample $\hat{\boldsymbol{Y}} \in \mathbb{R}^{F \times N}$ from the predicted distribution $\hat{f}(\boldsymbol{Y}_{\mathcal{F}})$ satisfies the coherence property. In particular, $\hat{f}$ denotes the density function (multi-variate) of the future values $\boldsymbol{Y}_{\mathcal{F}}$ conditioned on the historical data $\boldsymbol{X}_{\mathcal{H}}, \boldsymbol{Y}_{\mathcal{H}}$ and the future features $\boldsymbol{X}_{\mathcal{F}}$. We omit the conditioning from the expression for readability.

## 2.1 Related Work on Hierarchical Forecasting

**Coherent Point Forecasting:** As mentioned earlier, many existing coherent hierarchical forecasting methods rely on a two-stage reconciliation approach. More specifically, given non-coherent base forecasts $\hat{\boldsymbol{y}}_t$, reconciliation approaches aim to design a projection matrix $\boldsymbol{P} \in \mathbb{R}^{m \times n}$ that can project the base forecasts linearly into new leaf forecasts, which are then aggregated using $\boldsymbol{S}$ to obtain (coherent) revised forecasts $\tilde{\boldsymbol{y}}_t = \boldsymbol{S}\boldsymbol{P}\hat{\boldsymbol{y}}_t$. The post-processing is call reconciliation or on other words base forecasts are reconciled. Different hierarchical methods specify different ways to optimize for the $\boldsymbol{P}$ matrix. The naive Bottom-Up approach (Hyndman & Athanasopoulos, 2018) simply aggregates up from the base leaf predictions to obtain revised coherent forecasts. The MinT method (Wickramasuriya et al., 2019) computes $\boldsymbol{P}$ that obtains the minimum variance unbiased revised forecasts, assuming unbiased base forecasts. The ERM method from Ben Taieb & Koo (2019) optimizes $\boldsymbol{P}$ by directly performing empirical risk minimization over the mean squared forecasting errors. Several other criteria (Hyndman et al., 2011; Van Erven & Cugliari, 2015; Panagiotelis et al., 2020) for optimizing for $\boldsymbol{P}$ have also been proposed. Note that some of these reconciliation approaches like MinT can be used to generate confidence intervals by estimating the empirical covariance matrix of the base forecasts Hyndman & Athanasopoulos (2018).

**Coherent Probabilistic Forecasting:** The PERMBU method in Taieb et al. (2017) is a reconciliation based hierarchical approach for probabilistic forecasts. It starts with independent marginal probabilistic forecasts for all nodes, then uses samples from marginals at the leaf nodes, applies an empirical copula, and performs a mean reconciliation step to obtain revised (coherent) samples for the higher level nodes. Athanasopoulos et al. (2020) also discuss two approaches for coherent probabilistic forecasting: (i) using the empirical covariance matrix under the Gaussian assumption and (ii) using a non-parametric bootstrap method. The recent work of Rangapuram et al. (2021), Olivares et al. (2021) is a single-stage end-to-end method that uses deep neural networks to obtain coherent probabilistic hierarchical forecasts. Their approach is to use a neural-network based multivariate probabilistic forecasting model to jointly model all the time series and explictly incorporate a differentiable reconciliation step as part of model training, by using sampling and projection operations.

**Approximately Coherent Methods:** Some approximately-coherent hierarchical models have also been recently proposed, that mainly use the hierarchy information for improving prediction quality, but do not guarantee strict coherence and do not usually generate probabilistic predictions. Many of them (Mishchenko et al., 2019; Gleason, 2020; Han et al., 2021a;b; Paria et al., 2021) use regularization-based approaches to incorporate the hierarchy tree into the model via $\ell_2$ regularization. Kamarthi et al. (2022) impose approximate coherence on probabilistic forecasts via regularization of the output distribution.

# 3 Probabilistic top-down model

In many forecasting datasets (e.g. retail and traffic) it is often the case that the time-series closer to the top level have more well defined seasonal pattern and trends, and therefore are more predictable. However, as we go down the tree the time-series become sparser and more difficult to predict. This has been observed in prior works (Gross & Sohl, 1990; Athanasopoulos et al., 2009) and provides motivation for top-down modeling.

Our main contribution is a single shared *top-down proportions* model for predicting the future proportions of the children for any parent node in the tree [3]. Our model can easily outperform historical proportions and provide probabilistic forecasts, when combined with any probabilistic forecasting model for the root node.

**Top-down proportions model:** We propose a global top-down proportions model that predicts the future fractions/proportions according to which the future of any parent time-series disaggregates into its children time-series in the tree. This is based on the intuition that the proportions are more predictable given the history, as compared to the children time-series. Consider a *family* which is defined as a parent node $p$ along with its children $\mathcal{L}(p)$. For any child $c \in \mathcal{L}(p)$, define

$$a_{s,c} = \frac{y_{s,c}}{\sum_{j \in \mathcal{L}(p)} y_{s,j}}, \quad \text{for all } s \in [T]. \tag{1}$$

The matrix $\boldsymbol{A}(p) \in \mathbb{R}^{T \times C}$ denotes the proportions of the children over time, where $C := |\mathcal{L}(p)|$. We will drop the $p$ in braces when it is clear from context that we are dealing with a particular *family* $(p, \mathcal{L}(p))$. As in Section 2, we use $\boldsymbol{A}_{\mathcal{H}}$ and $\boldsymbol{A}_{\mathcal{F}}$ to denote the history and future proportions.

The task for our model is to predict the distribution of $\boldsymbol{A}_{\mathcal{F}}$ given historical proportions $\boldsymbol{A}_{\mathcal{H}}$, the parent's history $\boldsymbol{y}_{\mathcal{H}}^{(p)}$ and covariates $\boldsymbol{X}$. Note that for any $s \in [T]$, $\boldsymbol{a}_s \in \Delta^{C-1}$, where $\Delta^{d-1}$ denotes the $(d-1)$-dimensional simplex. Therefore, our predicted distribution should also be a distribution over the simplex for each row. Hence, we use the Dirichlet (Olkin & Rubin, 1964) family to model the output distribution for each row of the predicted proportions, as we will detail later.

*Architecture.* The two main architectural components that we use are (i) a LSTM based sequence to sequence (seq-2-seq) (Hochreiter & Schmidhuber, 1997) model to capture temporal structure and (ii) a multi-head self attention model (Vaswani et al., 2017) to capture the dependence across the children proportions. We first pass our inputs through the seq-2-seq model and obtain the decoder side output as follows,

$$\boldsymbol{D}_{\mathcal{F}} \leftarrow \texttt{seq2seq}\Big(\boldsymbol{A}_{\mathcal{H}}, \boldsymbol{y}_{\mathcal{H}}^{(p)}, \boldsymbol{X}, \boldsymbol{E}^{(\mathcal{L}(p))}\Big)$$

where $\boldsymbol{E}^{(\mathcal{L}(p))}$ are embeddings for the children nodes that are jointly trained. The decoder output $\boldsymbol{D}_{\mathcal{F}}$ has shape $F \times C \times r$, where $r$ denotes the output dimension of the decoder. Recall that we are working with $F$ steps of future as defined in Section 2. Note that each child proportion time-series is fed independently to the seq-2-seq model i.e $C$ is the batch dimension of the encoder-decoder, as shown in Figure 2. We then pass the decoder outputs through several layers of multi-headed self attention given by,

$$\boldsymbol{M}_{\mathcal{F}} \leftarrow \texttt{MultiHeadAtt}_{g,l}(\boldsymbol{D}_{\mathcal{F}}),$$

where $g$ denotes the number of attention heads and $l$ denotes the number of attention layers. Each attention layer is followed by a fully connected layer with ReLU activation and also equipped with a residual connection. Note that the attention is only applied across the third dimension i.e across the children. $\boldsymbol{M}_{\mathcal{F}}$ is of dimension $F \times C \times o$, where $o$ is the output dimension of the attention layers. We finally pass this through a linear layer of output size one, with exponential link function to get an output $\boldsymbol{B}_{\mathcal{F}} \in \mathbb{R}_+^{F \times C}$ that is the same dimension as that of $\boldsymbol{A}_{\mathcal{F}}$. We provide a full illustration of our model in Figure 2. Intuitively the self-attention models the dependencies between the proportions of the children of the same family.

*Loss Function.* Recall that the predicted proportions distributions $\hat{f}(\boldsymbol{A}_{\mathcal{F}})$ have to be over the simplex $\Delta^{C-1}$ for each family. Therefore we model it by the Dirichlet family. In fact our final model output for a family $\boldsymbol{B}_{\mathcal{F}}$ represents the parameters of predictive Dirichlet distributions. Specifically, we minimize the loss

$$\ell(\boldsymbol{B}_{\mathcal{F}}, \boldsymbol{A}_{\mathcal{F}}) = -\frac{1}{F} \sum_{s=t}^{t+F-1} \texttt{DirLL}(\boldsymbol{a}_s + \epsilon; \mathbf{b}_s), \tag{2}$$

---

[3]Note that Hyndman et al. (2011) state that top-down forecasting cannot be unbiased under the assumption that base forecasts for all levels are independent. We do not generate base forecasts first and then reconcile and this independence assumptions does not apply to our setting.

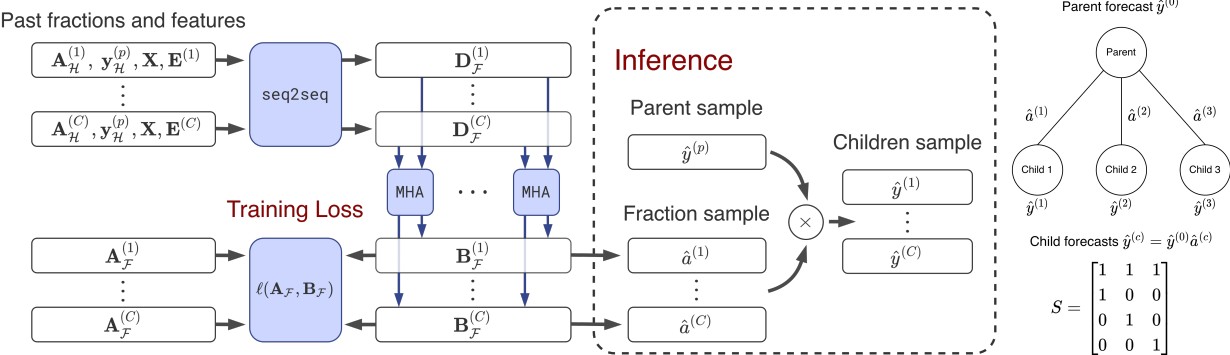

Figure 2: On the *left*, we provide a complete description of the training and inference architecture is shown in the figure. MHA denotes multi-head self attention layers and seq2seq denotes any sequence to sequence model such as LSTM based encoder-decoder models. The indices $\{1, \cdots, C\}$ are used to denote the indices of the children of parent node $p$. During inference the children fractions are sampled from the Dirichlet distribution and multiplied with the parent samples to yield the children samples. On the *right*, we show an example two level tree and the corresponding $S$ matrix.

where $\boldsymbol{a}_s$ is the proportion of the children nodes in the family at time $s$ as defined in Equation (1). $\texttt{DirLL}(\boldsymbol{a}; \boldsymbol{\beta})$ denotes the log-likelihood of Dirichlet distribution for target $\boldsymbol{a}$ and parameters $\boldsymbol{\beta}$.

$$\texttt{DirLL}(\boldsymbol{a}; \boldsymbol{\beta}) := \sum_i (\beta_i - 1)\log(a_i) - \log B(\boldsymbol{\beta}), \tag{3}$$

where $B(\boldsymbol{\beta})$ is the normalization constant. In Eq. (2), we add a small $\epsilon$ to avoid undefined values when the target proportion for some children are zero. Here, $B(\boldsymbol{\beta}) = \prod_{i=1}^{C} \Gamma(\beta_i)/\Gamma(\sum_i \beta_i)$ where $\Gamma(.)$ is the well-known Gamma function that is differentiable. In practice, we use Tensorflow Probability (Dillon et al., 2017) to optimize the above loss function.

*Training.* We train our top-down model with mini-batch gradient descent where each batch corresponds to different history and future time-intervals of the *same family*. For example, if the time batch-size is $b$ and we are given a family $(p, \mathcal{L}(p))$ the input proportions that are fed into the model are of shape $b \times H \times C$ and the output distribution parameters are of shape $b \times F \times C$, where $C = |\mathcal{L}(p)|$. Note that we only need to load all the time-series of a given family into a batch.

**Root probabilistic model:** Given the trained *top-down proportions* we can generate the distribution of proportions for any internal family of the hierarchy. Thus, if we obtain probabilistic forecasts for the root time-series we can achieve our original objective in Section 2. The advantage of our method is that any uni-variate/multi-variate probabilistic forecasting model (and advances there in) can be used for the root node. In this paper, we use Prophet (Taylor & Letham, 2018) for the root time series. The implementation details of the root probabilistic model can be found in Appendix C.

**Inference:** At inference we have to output a representation of the predicted cumulative distribution $\hat{F}(\boldsymbol{Y}_\mathcal{F})$ such that the samples are reconciled as in Section 2. We can achieve this by sampling from the predictive distribution of the two trained models. For ease of illustration, we will demonstrate the procedure for one time point $s \in \{t, \cdots, t + F - 1\}$.

We first sample $\hat{y}_s^{(r)}$ from the predictive distribution of the root node model. Then for every parent node (non leaf), $p$ in the tree we generate a sample $\hat{\boldsymbol{a}}_s^{(\mathcal{L}(p))}$ that represents a sample of the predicted children proportions for that family. The proportion samples and the root sample can be combined to form a reconciled forecast sample $\hat{\boldsymbol{y}}_s$. We can generate many such samples and then take empirical statistics to form the predictive distribution $\hat{f}(\boldsymbol{Y}_s)$, which is by definition reconciled.

# 4   Theoretical justification for the top-down approach

In this section, we theoretically analyze the advantage of the top-down approach over the bottom-up approach for hierarchical prediction in a simplified setting. Again, the intuition is that the root level time series is much less noisy and hence much easier to predict, and it is easier to predict proportions at the children nodes than the actual values themselves. As a result, combining the root level prediction with the proportions prediction actually yields a much better prediction for the children nodes. Consider a 2-level hierarchy of linear regression problem consisting of a single root node (indexed by 0) with $K$ children. For each time step $t \in [n]$, a global covariate $\mathbf{x}_t \in \mathbb{R}^d$ is independently drawn from a Gaussian distribution $\mathbf{x}_t \sim \mathcal{N}(0, \Sigma)$, and the value for each node is defined as follows:

- The value of the root node at time $t$ is $y_{t,0} = \theta_0^\top \mathbf{x}_t + \eta_t$, where $\eta_t \in \mathbb{R}$ is independent of $\mathbf{x}_t$, and satisfies $\mathbb{E}[\eta_t] = 0, \mathrm{Var}[\eta_t] = \sigma^2$.
- A random $K$-dimensional vector $\mathbf{a}_t \in \mathbb{R}^K$ is independently drawn from distribution $P$ such that $\mathbb{E}[a_{t,i}] = p_i$ and $\mathrm{Var}[a_{t,i}] = s_i$, where $a_{t,i}$ is the $i$-th coordinate of $\mathbf{a}_t$. For the $i$-th child node, the value of the node is defined as $a_{t,i} \cdot y_{t,0}$.

Notice that for the $i$-th child node, $\mathbb{E}[y_{t,i}|\mathbf{x}_t] = p_i \theta_0^\top \mathbf{x}_t$, and therefore the $i$-th child node follows from a linear model with coefficients $\theta_i := p_i \theta_0$.

Now we describe the bottom-up approach and top-down approach and analyze the expected excess risk of them respectively. In the bottom-up approach, we learn a separate linear predictor for each child node seprately. For the $i$-th child node, the ordinary least square (OLS) estimator is

$$\hat{\theta}_i^{\mathbf{b}} = \left(\sum_{t=1}^n \mathbf{x}_t \mathbf{x}_t^\top\right)^{-1} \sum_{t=1}^n \mathbf{x}_t y_{t,i},$$

and the prediction of the root node is simply the summation of all the children nodes.

In the top-down approach, a single OLS linear predictor is first learnt for the root node:

$$\hat{\theta}_0^{\mathbf{t}} = \left(\sum_{t=1}^n \mathbf{x}_t \mathbf{x}_t^\top\right)^{-1} \sum_{t=1}^n \mathbf{x}_t y_{t,0}$$

Then the proportion coefficient $\hat{p}_i, i \in [K]$ is learnt for each node separately as $\hat{p}_i = \frac{1}{n}\sum_{t=1}^n y_{t,i}/y_{t,0}$ and the final linear predictor for the $i$th child is $\hat{\theta}_i^{\mathbf{t}} = \hat{p}_i \hat{\theta}_0^{\mathbf{t}}$. Let us define the excess risk of an estimator $\hat{\theta}_i$ as $r(\hat{\theta}_i) = (\hat{\theta}_i - \theta_i)^\top \Sigma (\hat{\theta}_i - \theta_i)$. The expected excess risk of both approaches are summarized in the following theorem, proved in Appendix A.1.

**Theorem 4.1** (Expected excess risk comparison between top-down and bottom-up approaches)**.** *The total expected excess risk of the bottom-up approach for all the children nodes satisfies*

$$\sum_{i=1}^K \mathbb{E}[r(\hat{\theta}_i^{\mathbf{b}})] \geq \sum_{i=1}^K (s_i + p_i^2) \frac{d}{n-d-1} \sigma^2,$$

*and the total expected excess risk of the top-down approach satisfies*

$$\sum_{i=1}^K \mathbb{E}[r(\hat{\theta}_i^{\mathbf{t}})] = \frac{\sum_{i=1}^K s_i}{n} \theta_0^\top \Sigma \theta_0 + \left(\frac{\sum_{i=1}^K s_i}{n} + \sum_{i=1}^K p_i^2\right) \frac{d}{n-d-1} \sigma^2,$$

Applying the theorem to the case where the proportion distribution $\mathbf{a}_t$ is drawn from a uniform Dirichlet distribution, we show the excess risk of the traditional bottom-up approach is $\min(K, d)$ times bigger than our proposed top-down approach in the following corollary. A proof of the corollary can be found in Appendix A.2

**Corollary 4.2.** *Assuming that for each time-step $t \in [n]$, the proportion coefficient $\mathbf{a}_t$ is drawn from a $K$-dimensional Dirichlet distribution $Dir(\alpha)$ with $\alpha_i = \frac{1}{K}$ for all $i \in [K]$ and $\theta_0^\top \Sigma \theta_0 = \sigma^2$, then*

$$\frac{\mathbb{E}[\sum_{i=1}^K r(\hat{\theta}_i^{\mathbf{b}})]}{\mathbb{E}[\sum_{i=1}^K r(\hat{\theta}_i^{\mathbf{t}})]} = \Omega(\min(K, d)).$$

Table 1: Dataset features. The forecast horizon is denoted by $F$.

| Dataset | Total time series | Leaf time series | Levels | Observations | $F$ |
|---|---|---|---|---|---|
| M5 | 3060 | 3049 | 4 | 1913 | 7 days |
| Favorita | 4471 | 4100 | 4 | 1687 | 7 days |
| Tourism-L (Geo) | 111 | 76 | 4 | 228 | 12 months |
| Tourism-L (Trav) | 445 | 304 | 5 | 228 | 12 months |
| Traffic | 207 | 200 | 4 | 366 | 7 days |
| Labour | 57 | 32 | 4 | 514 | 8 months |
| Wiki | 199 | 150 | 5 | 366 | 7 days |

In Section 5, we show that a reasonable root level model combined with just historical fractions outperforms several state of the art methods on the M5 and Favorita datasets, thus conforming to our theoretical justification. Our learnt top-down model is a further improvement over the historical fractions.

## 5 Experiments

We implement our probabilistic top-down model in Tensorflow (Abadi et al., 2016) and compare against multiple baselines on six popular hierarchical time-series datasets.

**Datasets.** We experiment with two retail forecasting datasets, M5 (M5, 2020) and Favorita (Favorita, 2017), and all the datasets used in (Rangapuram et al., 2021): Tourism-L (Tourism, 2019; Wickramasuriya et al., 2019) which is a dataset consisting of tourist count data. Labour (of Statistics, 2020), consisting of monthly employment data, Traffic (Cuturi, 2011) (consisting of daily occupancy rates of cars on freeways), and Wiki (Wiki, 2017) (consisting of daily views on Wikipedia articles). For M5 and Favorita we use the product hierarchy. For Tourism-L we benchmark on both the (Geo)graphic and (Trav)el history based hierarchy. More details about the dataset and the features used for each dataset can be found in Appendix B and Table 1. Note that for the sake of reproducibility, except for the additional M5 and Favorita datasets, the datasets and experimental setup are largely identical to that in (Rangapuram et al., 2021) with an increased horizon for traffic and wiki datasets. In (Rangapuram et al., 2021), the prediction window for these datasets were chosen to be only 1 time-step which is extremely small; moreover on traffic the prediction window only includes the day Dec 31st which is atypical especially because the dataset includes only an year of daily data. Therefore we decided to increase the validation and test size to 7. Favorita and M5 are among the largest among the popularly used public hierarchical forecasting datasets and therefore are ideal for benchmarking methods that are both scalable and accurate.

**Model details.** For our proportions model, we set the validation split to be of the same size as the test set, and immediately preceding it in time, which is standard (Rangapuram et al., 2018). We tune several hyper-parameters using the validation set loss, the details of which are provided in Appendix E. Then we use the best hyper-parameter model to predict the Dirichlet parameters for proportions in the test set. We separately tune and train a Prophet model (Taylor & Letham, 2018) on the training set for the root node time-series model. (please refer to Appendix C for more details, including our results with other root models). We then combine the predicted samples from the proportions model and root models in order to generate the predicted quantiles for all time-series in the hierarchy, as detailed in Section 3. We refer to our overall model as `TDProb`.

**Baselines.** We compare our model to the following coherent hierarchical forecasting baselines: (i) `Hier-E2E` (Rangapuram et al., 2021) is an end-to-end deep-learning approach for coherent probabilistic forecasts. (ii) `PERMBU-MINT` (Taieb et al., 2017) is a copula based reconciliation approach for producing probabilistic hierarchical forecasts. (iii) `DeepAR-BU` uses a recent deep, probabilistic model DeepAR (Salinas et al., 2020) at the leaf nodes, samples from the leaf node distributions and performs bottom-up aggregation on the samples to obtain coherent probabilistic predictions at all nodes. Similarly, `DeepSSM-BU` is the bottom-up model that uses the deep-state space model Rangapuram et al. (2018) as the leaf nodes forecaster. (iv) `Prophet-Historical` is a top-down model based on using Prophet at the root node, and relying on historical fractions to generate predictions at lower-level nodes. (v) `ETS-BU`, `ARIMA-BU` are bottom up approaches, where base forecasts produced using ETS and ARIMA models are aggregated to produce aggregate level predictions. (vi) `ETS-MINT-OLS`, `ETS-MINT-SHR`, `ARIMA-MINT-OLS`, `ARIMA-MINT-SHR` (Wickramasuriya et al., 2019) are

Table 2: Normalized CRPS scores for all the datasets introduced in Sec 5. We average the deep learning based methods over 10 independent runs. The rest of the methods had very little variance. We report the corresponding standard error and only bold numbers that are the statistically significantly better than the rest. The second best numbers in each column are italicized. Note: The official implementation (Hyndman et al., 2015) of the baselines PERMBU-MINT, ETS-MINT-SHR, ETS-ERM, ARIMA-MINT-SHR, ARIMA-MINT-OLS, and ARIMA-ERM returned invalid values (NaNs) either for the Favorita or the M5 dataset - our largest datasets (possibly because of numerical issues), and were omitted from the table. These baselines appear for every other dataset. We report only mean metrics across all hierarchical levels for Labour, Traffic, Wiki2, and Tourism for lack of space. The full set of level-wise metrics can be found in Appendix D.

| M5 | Model Type | L0 | L1 | L2 | L3 | Mean |
|---|---|---|---|---|---|---|
| TDProb | | *0.0227* ± 0.0001 | **0.0273** ± 0.0004 | **0.0304** ± 0.0004 | **0.1992** ± 0.0006 | **0.0699** ± 0.0002 |
| Prophet + historical fractions | | 0.0227 ± 0.0 | 0.0289 ± 0.0 | 0.0409 ± 0.0 | 0.2612 ± 0.0 | 0.0884 ± 0.0 |
| Hier-E2E | Probabilistic | 0.1143 ± 0.0039 | 0.1109 ± 0.0039 | 0.1175 ± 0.0039 | 0.2862 ± 0.003 | 0.1572 ± 0.0035 |
| DeepAR-BU | | 0.0421 ± 0.0027 | 0.0442 ± 0.0023 | 0.0497 ± 0.0020 | *0.2092* ± 0.0003 | 0.0863 ± 0.0017 |
| DeepSSM-BU | | 0.0294 | 0.0331 | 0.0420 | 0.2898 | 0.0986 |
| PERMBU-MINT | | **0.0224** | *0.0281* | *0.0316* | 0.2147 | *0.0742* |
| ETS-BU | | 0.0386 | 0.0490 | 0.0536 | 0.2905 | 0.1079 |
| ETS-MINT-OLS | | 0.0356 | 0.0457 | 0.0508 | 0.2853 | 0.1043 |
| ETS-MINT-SHR | | 0.0408 | 0.0498 | 0.0539 | 0.2856 | 0.1075 |
| ETS-ERM | Point | 0.3491 | 0.3502 | 0.3676 | 0.9406 | 0.5019 |
| ARIMA-BU | | 0.1116 | 0.1127 | 0.1162 | 0.3006 | 0.1602 |
| ARIMA-MINT-SHR | | 0.0671 | 0.0729 | 0.0752 | 0.2896 | 0.1262 |
| ARIMA-ERM | | 0.0590 | 0.0616 | 0.0731 | 0.4043 | 0.1495 |

| Favorita | Model Type | L0 | L1 | L2 | L3 | Mean |
|---|---|---|---|---|---|---|
| TDProb | | **0.031** ± 0.0005 | **0.0503** ± 0.0005 | **0.0711** ± 0.0006 | **0.1478** ± 0.0009 | **0.0751** ± 0.0005 |
| Prophet + historical fractions | | 0.031 ± 0.0 | *0.061* ± 0.0 | *0.103* ± 0.0 | 0.224 ± 0.0 | *0.104* ± 0.0 |
| Hier-E2E | Probabilistic | *0.0635* ± 0.0027 | 0.0944 ± 0.0023 | 0.1427 ± 0.0021 | 0.274 ± 0.0021 | 0.1437 ± 0.002 |
| DeepAR-BU | | 0.0854 ± 0.0068 | 0.0920 ± 0.0062 | *0.1018* ± 0.0057 | 0.1744 ± 0.0054 | 0.1134 ± 0.0059 |
| DeepSSM-BU | | 0.0979 | 0.1046 | 0.1372 | 0.3602 | 0.1750 |
| ETS-BU | | 0.0987 | 0.1061 | 0.1206 | *0.1502* | 0.1189 |
| ETS-MINT-OLS | Point | 0.106 | 0.1135 | 0.1355 | 0.1669 | 0.1305 |
| ARIMA-BU | | 0.0856 | *0.0962* | 0.1205 | 0.2199 | 0.1306 |
| ARIMA-MINT-OLS | | 0.1266 | 0.1444 | 0.1504 | 0.2084 | 0.1574 |

| Mean metrics | Labour | Traffic | Wiki2 | Tourism |
|---|---|---|---|---|
| TDProb | **0.0318** ± 0.0005 | *0.0575* ± 0.0006 | **0.2662** ± 0.0005 | **0.1372** |
| Prophet + historical fractions | *0.033* ± 0.0 | 0.1111 ± 0.0 | *0.2744* ± 0.0 | 0.1803 |
| Hier-E2E | 0.0340 ± 0.0088 | **0.0506** ± 0.0011 | *0.2769* ± 0.004 | 0.1520 |
| DeepAR-BU | 0.0401 ± 0.0024 | 0.0840 ± 0.0023 | 0.3637 ± 0.0045 | *0.1438* |
| DeepSSM-BU | 0.0531 | 0.0583 | 0.3240 | 0.1602 |
| Best of Rest | 0.0393 ± 0.0002 | 0.1363 | 0.4418 | 0.1609 |

MinT based reconciliation approaches on base forecasts produced using ETS and ARIMA. SHR corresponds to the covariance matrix with a shrinkage operator, and OLS denotes a diagonal covariance matrix. (vii) ETS-ERM, ARIMA-ERM (Ben Taieb & Koo, 2019) are ERM based methods applied to the base forecasts from ETS and ARIMA models. Another method SHARQ (Han et al., 2021a), consists of an independent deep model for each node in the hierarchy, and is trained sequentially one node at a time. The official implementation[4] did not scale to our datasets, as it is infeasible to train a separate model for each node.

We use the public implementation released in Rangapuram et al. (2021), and the public GluonTS forecasting library Alexandrov et al. (2019) for running all the above baselines. In Table 2 we indicate the models that can produce coherent probabilistic forecasts versus coherent point forecasts. For the methods that produce point forecasts, we use the point predictions for calculating the quantile losses for all quantiles in the CRPS expression. This is the same convention that was followed in the prior work (Rangapuram et al., 2021). We would further like to note that reconciliation methods such as MinT can be modified to produce coherent covariant matrices that enables probabilistic forecasts (Hyndman & Athanasopoulos, 2018); however we do not compare with this approach since PERMBU-MINT (which is included in our benchmarks) is an improvement over this method(Taieb et al., 2017). *Note that our baselines are a* **strict superset** *of baselines used by Rangapuram et al. (2021).*

**Evaluation.** We evaluate forecasting accuracy using the continuous ranked probability score (CRPS). The CRPS (Gneiting & Raftery, 2007) is minimized when the predicted quantiles match the true data distribution.

---

[4]https://github.com/aaronhan223/htsf

Table 3: We perform an ablation study of our model on the Wiki2 dataset that compares different root models. We show the results when our proportions model is combined with two other root level models. We present results with DeepSSM on L0 with out proportions model (`TDProbDeepSSM`) and also ARIMA + our proportions models (`TDProbAR`).

| Wiki2 | Mean |
|---|---|
| TDProb | *0.2662* ± 0.0005 |
| TDProbDeepSSM | **0.2504** ± 0.0020 |
| TDProbAR | 0.2674 ± 0.0031 |
| Hier-E2E (Best of Rest) | 0.2769 ± 0.004 |

This is the standard metric used to benchmark probabilistic forecasting in numerous papers (Rangapuram et al., 2018; 2021; Taieb et al., 2017).

Denote the $F$ step $q$-quantile prediction for time series $i$ by $\hat{\boldsymbol{Q}}_{\mathcal{F}}^{(i)}(q) \in \mathbb{R}^F$. $\hat{\boldsymbol{Q}}_s^{(i)}(q)$ denotes the $q$-th quantile prediction for the $s$-th future time-step for time-series $i$, where $s \in [F]$. Then the CRPS loss is:

$$\text{CRPS}(\hat{\boldsymbol{Q}}_{\mathcal{F}}^{(i)}(q), \boldsymbol{Y}_{\mathcal{F}}^{(i)}) = \frac{1}{F} \sum_{s \in [F]} \int_0^1 2(\mathbb{I}[\boldsymbol{Y}_s^{(i)} \leq \hat{\boldsymbol{Q}}_s^{(i)}(q)] - q)(\hat{\boldsymbol{Q}}_s^{(i)}(q) - \hat{\boldsymbol{Y}}_s^{(i)})dq. \tag{4}$$

We report the CRPS scores of the prediction for each individual level of the hierarchy. Similar to Rangapuram et al. (2021), we also normalize the CRPS scores at each level, by the absolute sum of the true values of all the nodes of that level. We also report the mean of the level-wise scores denoted by *Mean* in Table 2.

We present level-wise performance of all methods on M5 and Favorita, as well as average performance on the other datasets (the full level-wise metrics on all datasets can be seen in Appendix D). In these tables, we highlight in bold numbers that are statistically significantly better than the rest. The second best numbers in each column are italicized. The deep learning based methods are averaged over 10 runs while other methods had very little variance. We also present results with different root level models when combined with our top-down proportions model.

**M5:** We see that overall in the mean column, `TDProb` performs the best (around 6% better than the best baseline `PERMBU-MINT` ). In fact, `TDProb` performs the best for all levels except L0, where `PERMBU-MINT` achieves the best CRPS (closely followed by `TDProb` ). This shows that our proportions model can be coupled with off-the-shelf univariate forecasting models to achieve state-of-the-art performance. We hypothesize that `Hier-E2E` does not work well on these larger datasets because the DeepVAR model needs to be applied to thousands of time-series, which leads to a prohibitive size of the fully connected input layer and a hard joint optimization problem. In Table 8 we provide similar results on a longer horizon forecasting task where $\tau = 35$.

**Favorita:** In all rows, `TDProb` outperforms the other models by a large margin, resulting in a 27% better mean performance than the best baseline. Interestingly, the simple `Prophet-Historical` model is the second best in the mean column, outperforming both `Hier-E2E` and `DeepAR-BU`. This suggests that even a naive top-down model coupled with a strong root-level univariate model can sometimes outperform deep learning-based bottom-up and reconciliation models. Furthermore, using a sophisticated, probabilistic, proportions model such as `TDProb` significantly improves top-down methods.

We also report WAPE/ NRMSE results on p50 (median) predictions in Table 6. We see that our model provides a 18.7% lower NRMSE over the best baseline in terms of the mean across all levels. We also report CRPS scores for a longer horizon task ($\tau = 35$) in Table 7 where we see a gain of 35% over the best baseline in the mean metrics.

**Other Datasets:** Table 2 presents mean CRPS scores on Labor, Traffic, Tourism and Wiki datasets. Except for Traffic (where `Hier-E2E` is better), we can see that `TDProb` outperforms all the other baselines in all other datasets. In Labour and Wiki, in addition to `TDProb` the naive top-down `Prophet-Historical` model again performs better than the other bottom-up and reconciliation methods. In Tourism-L dataset, the `DeepAR-BU` model performs second-best, followed by `Hier-E2E`. We provide more detailed results for all these datasets in Appendix D.

**Root Model:** In Table 3 we dive deeper into the role of the root-level model using the Wiki2 dataset. The first row is our actual `TDProb` model. Then, we present results with DeepSSM on L0 with our proportions model (`TDProbDeepSSM`) and also ARIMA + our proportions models (`TDProbAR`). `TDProbDeepSSM` is chosen because on this dataset DeepSSM performs exceedingly well on the top-level (see Appendix D) and this indeed obtains the best result overall. `TDProbAR` is chosen as ARIMA is one of the most commonly used and simplest to implement models. We show that our proportions model can be combined with various reasonable root models in order to yield significantly better results than the baselines. More detailed results showing the CRPS at all levels are provided in Table 4 in the appendix.

## 6 Conclusion

In this paper, we proposed a probabilistic top-down based hierarchical forecasting approach, that obtains coherent, probabilistic forecasts without the need for a separate reconciliation stage. Our approach is built around a novel deep-learning model for learning the distribution of proportions according to which a parent time series is disaggregated into its children time series. Our model is flexible enough to be coupled with any univariate probabilistic forecasting method of choice for the root time series. We show in empirical evaluation on several public datasets, that our model obtains state-of-the-art results compared to previous methods.

For future work, we plan to explore extending our approach to handle more complex hierarchical structural constraints, beyond trees. We would also like to note that currently our theoretical justification only applies to learning historical proportions; it would be interesting to extend it to predicted future proportions. It should be noted that our work does not directly apply to approximately coherent data that can appear in differentially private hierarchical time series datasets.

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

## A  Proofs

### A.1  Proof of Theorem 4.1

We prove the claims about the excess risk of top-down and bottom-up approaches in the following two sections. Recall that ordinary least square (OLS) estimator $\hat{\theta} = \left(\sum_{i=1}^n \mathbf{x}_i \mathbf{x}_i^\top\right)^{-1} \sum_{i=1}^n \mathbf{x}_i y_i$. The population squared error of a linear predictor is defined as $(\hat{\theta} - \theta)^\top \Sigma (\hat{\theta} - \theta)$, which is also known as excess risk.

### A.1.1  Excess risk of the top down approach

For the root node, the OLS predictor is written as

$$\hat{\theta}_0 = \left(\sum_{t=1}^n \mathbf{x}_t \mathbf{x}_t^\top\right)^{-1} \sum_{t=1}^n \mathbf{x}_t y_{t,0},$$

and the expected excess risk is

$$
\begin{aligned}
&\mathbb{E}[(\hat{\theta}_0 - \theta_0)^\top \Sigma (\hat{\theta}_0 - \theta_0)] \\
&\overset{(a)}{=} \mathbb{E}\left[\sigma^2 \sum_{t=1}^n \mathbf{x}_t^\top \left(\sum_{r=1}^n \mathbf{x}_t \mathbf{x}_t^\top\right)^{-1} \Sigma \left(\sum_{r=1}^n \mathbf{x}_t \mathbf{x}_t^\top\right)^{-1} \mathbf{x}_t\right] \\
&\overset{(b)}{=} \mathrm{Tr}\left[\mathbb{E}\left[\sigma^2 \left(\sum_{t=1}^n \mathbf{x}_t \mathbf{x}_t^\top\right)^{-1} \Sigma\right]\right] \\
&\overset{(c)}{=} \sigma^2 d/(n - d - 1)
\end{aligned}
\tag{5}
$$

where equation (a) holds by expanding $y_{i,0} = \mathbf{x}_i^\top \theta_0 + \eta_i$ and the fact that $\eta_i$ is independent of $\mathbf{x}_i$, equation (b) holds by the property of trace, and equation (c) follows from the mean of the inverse-Wishart distribution.

For each children node, we learn the proportion coefficient with

$$\hat{p}_i = \frac{1}{n} \sum_{t=1}^{n} \frac{y_{t,i}}{y_{t,0}}.$$

Notice that

$$
\begin{aligned}
\mathrm{Var}[\hat{p}_i] &= \frac{1}{n} \mathrm{Var}\left[\frac{y_{1,i}}{y_{1,0}}\right] \\
&= \frac{1}{n} \mathrm{Var}[a_i] \\
&= s_i/n.
\end{aligned}
\tag{6}
$$

Recall that the optimal linear predictor of the $i$-th child node is $p_i \theta_0$. Therefore, the expected excess risk of the top down predictor is

$$
\begin{aligned}
&\mathbb{E}\left[(\hat{p}_i \hat{\theta}_0 - p_i \theta_0)^\top \Sigma (\hat{p}_i \hat{\theta}_0 - p_i \theta_0)\right] \\
&= \mathbb{E}\left[(\hat{p}_i \hat{\theta}_0 - p_i \hat{\theta}_0 + p_i \hat{\theta}_0 - p_i \theta_0)^\top \Sigma (\hat{p}_i \hat{\theta}_0 - p_i \hat{\theta}_0 + p_i \hat{\theta}_0 - p_i \theta_0)\right] \\
&= \mathbb{E}\left[(\hat{p}_i - p_i)^2 \hat{\theta}_0^\top \Sigma \hat{\theta}_0 + p_i^2 (\hat{\theta}_0 - \theta_0)^\top \Sigma (\hat{\theta}_0 - \theta_0)\right] \\
&\stackrel{(a)}{=} \frac{1}{n} s_i \theta_0^\top \Sigma \theta_0 + \left(\frac{1}{n} s_i + p_i^2\right) \frac{d}{n-d-1} \sigma^2,
\end{aligned}
$$

where we have applied Equation 6 and Equation 5 in equality (a). Taking summation over all the children, we get the total excess risk equals

$$\frac{\sum_{i=1}^{K} s_i}{n} \theta_0^\top \Sigma \theta_0 + \left(\frac{\sum_{i=1}^{K} s_i}{n} + \sum_{i=1}^{K} p_i^2\right) \frac{d}{n-d-1} \sigma^2$$

### A.1.2 Excess risk of the bottom up approach

For the $i$-th child node, the OLS estimator is

$$\hat{\theta}_i = \left(\sum_{t=1}^{n} \mathbf{x}_t \mathbf{x}_t^\top\right)^{-1} \sum_{t=1}^{n} \mathbf{x}_t y_{t,i}.$$

Recall that the best linear predictor of the $i$-th child node is $p_i \theta_0$ The excess risk is

$$
\begin{aligned}
&\mathbb{E}(\hat{\theta}_i - p_i \theta_0)^\top \Sigma (\hat{\theta}_i - p_i \theta_0) \\
&= \mathbb{E}\left[((\hat{\theta}_i - p_i \hat{\theta}_0) + (p_i \hat{\theta}_0 - p_i \theta_0))^\top \Sigma ((\hat{\theta}_i - p_i \hat{\theta}_0) + (p_i \hat{\theta}_0 - p_i \theta_0))\right]
\end{aligned}
$$

Notice that the cross term has 0 expectation as

$$
\begin{aligned}
&\mathbb{E}[(\hat{\theta}_i - p_i \hat{\theta}_0)^\top \Sigma (p_i \hat{\theta}_0 - p_i \theta_0)] \\
&\stackrel{(a)}{=} \mathbb{E}\left[\mathbb{E}_{\mathbf{a}}\left[\sum_{t=1}^{n}(a_{t,i} y_{t,0} - p_i y_{t,0})\mathbf{x}_t^\top \left(\sum_{t=1}^{n} \mathbf{x}_t \mathbf{x}_t^\top\right)^{-1} \Sigma (p_i \hat{\theta}_0 - p_i \theta_0)\right]\right] \\
&\stackrel{(b)}{=} 0,
\end{aligned}
$$

where the first equality holds by the definition of node $i$-th value $y_{t,i}$. Therefore, it holds that

$$\mathbb{E}\left[((\hat{\theta}_i - p_i\hat{\theta}_0) + (p_i\hat{\theta}_0 - p_i\theta_0))^\top \Sigma((\hat{\theta}_i - p_i\hat{\theta}_0) + (p_i\hat{\theta}_0 - p_i\theta_0))\right]$$

$$= \mathbb{E}\left[(\hat{\theta}_i - p_i\hat{\theta}_0)^\top \Sigma(\hat{\theta}_i - p_i\hat{\theta}_0)\right] + p_i^2 \frac{d}{n-d-1}\sigma^2$$

$$= \mathbb{E}\,\mathrm{Tr}\left[\sum_{j=1}^n (a_{j,i} - p_i)^2 y_{t,0}^2 \mathbf{x}_j\mathbf{x}_j^\top \left(\sum_{t=1}^n \mathbf{x}_t\mathbf{x}_t^\top\right)^{-1} \Sigma\left(\sum_{t=1}^n \mathbf{x}_t\mathbf{x}_t^\top\right)^{-1}\right] + p_i^2 \frac{d}{n-d-1}\sigma^2$$

$$= s_i \mathbb{E}\,\mathrm{Tr}\left[\sum_{j=1}^n ((\theta_0^\top\mathbf{x}_j)^2 + \eta_j^2)\mathbf{x}_j\mathbf{x}_j^\top \left(\sum_{t=1}^n \mathbf{x}_t\mathbf{x}_t^\top\right)^{-1} \Sigma\left(\sum_{t=1}^n \mathbf{x}_t\mathbf{x}_t^\top\right)^{-1}\right] + p_i^2 \frac{d}{n-d-1}\sigma^2$$

$$\overset{(a)}{\geq} s_i\sigma^2 \mathbb{E}\,\mathrm{Tr}\left[\left(\sum_{j=1}^n \mathbf{x}_j\mathbf{x}_j^\top\right)\left(\sum \mathbf{x}_i\mathbf{x}_i^\top\right)^{-1}\Sigma\left(\sum \mathbf{x}_i\mathbf{x}_i^\top\right)^{-1}\right] + p_i^2 \frac{d}{n-d-1}\sigma^2$$

$$\overset{(b)}{=} (s_i + p_i^2)\sigma^2 \frac{d}{n-d-1},$$

where inequality (a) holds since $(\theta_0^\top\mathbf{x}_j)^2$ term is non-negative, equality (b) holds by the property of inverse-Wishart distribution. Taking summation over all the children, we get the total excess risk is lower bounded by

$$\sum_{i=1}^K (s_i + p_i^2)\frac{d}{n-d-1}\sigma^2$$

This concludes the proof.

## A.2 Proof of Corollary 4.2

In this section, we apply Theorem 4.1 to Dirichlet distribution to show that the excess risk of bottom-up approach is $\min(d, K)$ times higher than top-down approach for a natural setting.

Recall that a random vector $\mathbf{a}$ drawn from a $K$-dimensional Dirichlet distribution $\mathrm{Dir}(\alpha)$ with parameters $\alpha$ has mean $\mathbb{E}[\mathbf{a}] = \frac{1}{\sum_{i=1}^K \alpha_i}\alpha$, and the variance $\mathrm{Var}[a_i] = \frac{\alpha_i(1-\alpha_i)}{\sum_{i=1}^K \alpha_i + 1}$. Let $\alpha_i = \frac{1}{K}$ for all $i \in [K]$, $\theta_0^\top\Sigma\theta_0 = \sigma^2$. The total excess risk of the top-down approach is

$$\mathbb{E}\left[\sum_{i=1}^K r(\hat{\theta}_i^{\mathbf{t}})\right] = \frac{\sum_{i=1}^K s_i}{n}\theta_0^\top\Sigma\theta_0 + \left(\frac{\sum_{i=1}^K s_i}{n} + \sum_{i=1}^K p_i^2\right)\frac{d}{n-d-1}\sigma^2$$

$$= \left(\frac{1-1/K}{2n} + \left(\frac{1-1/K}{2n} + \frac{1}{K}\right)\frac{d}{n-d-1}\right)\sigma^2.$$

The total excess risk of the bottom-up approach is lower bounded by

$$\mathbb{E}\left[\sum_{i=1}^K r(\hat{\theta}_i^{\mathbf{b}})\right] = (s_i + p_i^2)\frac{d}{n-d-1}\sigma^2$$

$$= \left(\frac{1-1/K}{2} + \frac{1}{K}\right)\frac{d}{n-d-1}\sigma^2$$

Now assuming that $n \geq 2d$, the top-down approach has expected risk $\mathbb{E}[\sum_{i=1}^K r(\hat{\theta}_i^{\mathbf{t}})] = O(\frac{1}{n} + \frac{d}{nK})$, and the bottom-up approach has expected risk $\mathbb{E}[\sum_{i=1}^K r(\hat{\theta}_i^{\mathbf{t}})] = \Omega(\frac{d}{n})$. Therefore, it holds that

$$\frac{\mathbb{E}[\sum_{i=1}^K r(\hat{\theta}_i^{\mathbf{b}})]}{\mathbb{E}[\sum_{i=1}^K r(\hat{\theta}_i^{\mathbf{t}})]} = \Omega(\min(d, K))$$

Table 4: Normalized CRPS scores on Tourism-L, Labour, Traffic, and Wiki2. We average the deep learning based methods over 10 independent runs. The rest of the methods had very little variance. We report the corresponding standard error and only bold the numbers that are significantly better than the rest. The second best numbers in each column are italicized. We also report the mean performance across all levels in the corresponding column. Rangapuram et al. (2018) have already distilled the best out of all considered baselines in their Table 4. For ease of comparison, we restate those numbers. On the wiki dataset, we also report numbers for our probabilistic top-down model combined with the L0 prediction of the model with the best L0 model (DeepSSM) and also TDProb with an ARIMA model on the top-level.

| Tourism | L0 | L1 (Geo) | L2 (Geo) | L3 (Geo) | L1 (Trav) | L2 (Trav) | L3 (Trav) | L4 (Trav) | Mean |
|---|---|---|---|---|---|---|---|---|---|
| TDProb | $0.0299 \pm 0.0003$ | $\mathbf{0.0781} \pm 0.0013$ | $\mathbf{0.1177} \pm 0.0012$ | $0.1642 \pm 0.0014$ | $0.0946 \pm 0.0035$ | $\mathbf{0.141} \pm 0.0027$ | $\mathbf{0.2023} \pm 0.0024$ | $0.2698 \pm 0.0023$ | $\mathbf{0.1372}$ |
| Prophet + historical fractions | $0.0299 \pm 0.0$ | $0.0869 \pm 0.0$ | $0.1636 \pm 0.0$ | $0.2247 \pm 0.0$ | $0.1128 \pm 0.0$ | $0.1860 \pm 0.0$ | $0.2760 \pm 0.0$ | $0.3632 \pm 0.0$ | $0.1803$ |
| Hier-E2E | $0.0810 \pm 0.0053$ | $0.1030 \pm 0.0030$ | $0.1361 \pm 0.0024$ | $0.1752 \pm 0.0026$ | $0.1027 \pm 0.0062$ | $\mathbf{0.1403} \pm 0.0047$ | $0.2050 \pm 0.0028$ | $0.2727 \pm 0.0017$ | $0.1520$ |
| DeepAR-BU | $0.0640 \pm 0.0034$ | $\mathbf{0.0784} \pm 0.0020$ | $\mathbf{0.1155} \pm 0.0012$ | $\mathbf{0.1561} \pm 0.0007$ | $0.123 \pm 0.006$ | $0.146 \pm 0.003$ | $\mathbf{0.203} \pm 0.002$ | $\mathbf{0.265} \pm 0.001$ | $0.1438$ |
| DeepSSM-BU | $0.0407$ | $0.0835$ | $0.1311$ | $0.1844$ | $0.0884$ | $0.1540$ | $0.2531$ | $0.3469$ | $0.1602$ |
| Best of Rest | $0.0438$ | $0.0816$ | $0.1433$ | $0.2036$ | $\mathbf{0.0830}$ | $0.1479$ | $0.2437$ | $0.3406$ | $0.1609$ |

| Labour | L0 | L1 | L2 | L3 | Mean |
|---|---|---|---|---|---|
| TDProb | $\mathbf{0.0299} \pm 0.0$ | $\mathbf{0.0285} \pm 0.0007$ | $\mathbf{0.0283} \pm 0.0007$ | $\mathbf{0.0406} \pm 0.0008$ | $\mathbf{0.0318} \pm 0.0005$ |
| Prophet + historical fractions | $\mathbf{0.0299} \pm 0.0$ | $0.0299 \pm 0.0$ | $0.0306 \pm 0.0$ | $0.043 \pm 0.0$ | $0.033 \pm 0.0$ |
| Hier-E2E | $0.0311 \pm 0.0120$ | $0.0336 \pm 0.0089$ | $0.0336 \pm 0.0082$ | $\mathbf{0.0378} \pm 0.0060$ | $0.0340 \pm 0.0088$ |
| DeepAR-BU | $0.0347 \pm 0.0027$ | $0.0433 \pm 0.0025$ | $0.0424 \pm 0.0024$ | $\mathbf{0.0400} \pm 0.0022$ | $0.0401 \pm 0.0024$ |
| DeepSSM-BU | $0.0512$ | $0.0569$ | $0.0540$ | $0.0500$ | $0.0531$ |
| Best of Rest | $0.0406 \pm 0.0002$ | $0.0389 \pm 0.0002$ | $0.0382 \pm 0.0002$ | $\mathbf{0.0397} \pm 0.0003$ | $0.0393 \pm 0.0002$ |

| Traffic ($F = 7$) | L0 | L1 | L2 | L3 | Mean |
|---|---|---|---|---|---|
| TDProb | $0.026 \pm 0.0003$ | $0.0282 \pm 0.0003$ | $0.034 \pm 0.0005$ | $0.1419 \pm 0.002$ | $0.0575 \pm 0.0006$ |
| Prophet + historical fractions | $0.0828 \pm 0.0013$ | $0.0863 \pm 0.0$ | $0.0903 \pm 0.0$ | $0.1821 \pm 0.0001$ | $0.1111 \pm 0.0$ |
| Hier-E2E | $0.0245 \pm 0.0011$ | $\mathbf{0.0268} \pm 0.001$ | $\mathbf{0.0307} \pm 0.0011$ | $\mathbf{0.1206} \pm 0.0019$ | $\mathbf{0.0506} \pm 0.0011$ |
| DeepAR-BU | $0.0659 \pm 0.0032$ | $0.0638 \pm 0.0029$ | $0.0625 \pm 0.0025$ | $0.1439 \pm 0.0006$ | $0.0840 \pm 0.0023$ |
| DeepSSM-BU | $\mathbf{0.0208}$ | $0.0303$ | $0.0356$ | $0.1465$ | $0.0583$ |
| ARIMA-ERM (Best of rest) | $0.073$ | $0.0869$ | $0.092$ | $0.2932$ | $0.1363$ |

| Wiki2 ($F = 7$) | L0 | L1 | L2 | L3 | L4 | Mean |
|---|---|---|---|---|---|---|
| TDProb | $0.0939 \pm 0.0018$ | $0.2205 \pm 0.0024$ | $0.2878 \pm 0.0024$ | $0.2968 \pm 0.0025$ | $0.432 \pm 0.0035$ | $0.2662 \pm 0.0005$ |
| TDProbDeepSSM | $\mathbf{0.0552} \pm 0.001$ | $0.2063 \pm 0.0037$ | $\mathbf{0.2793} \pm 0.0026$ | $\mathbf{0.2894} \pm 0.0026$ | $0.4235 \pm 0.0018$ | $\mathbf{0.2504} \pm 0.0020$ |
| TDProbAR | $0.1026 \pm 0.002$ | $0.2196 \pm 0.0028$ | $0.2907 \pm 0.0046$ | $0.2983 \pm 0.0043$ | $0.4257 \pm 0.0047$ | $0.2674 \pm 0.0031$ |
| Prophet + historical fractions | $0.0939 \pm 0.0$ | $\mathbf{0.1968} \pm 0.0$ | $0.3017 \pm 0.0$ | $0.3104 \pm 0.0$ | $0.4696 \pm 0.0$ | $0.2744 \pm 0.0$ |
| Hier-E2E | $0.133 \pm 0.0102$ | $0.2094 \pm 0.0057$ | $0.2942 \pm 0.0032$ | $0.3057 \pm 0.0031$ | $0.4421 \pm 0.0016$ | $0.2769 \pm 0.004$ |
| DeepAR-BU | $0.3399 \pm 0.0070$ | $0.3443 \pm 0.0056$ | $0.3625 \pm 0.0042$ | $0.3654 \pm 0.0039$ | $\mathbf{0.4065} \pm 0.0022$ | $0.3637 \pm 0.0045$ |
| DeepSSM-BU | $\mathbf{0.0552}$ | $0.2344$ | $0.3771$ | $0.3847$ | $0.5683$ | $0.3240$ |
| ETS-ERM (Best of Rest) | $0.3719$ | $0.4018$ | $0.438$ | $0.4482$ | $0.5491$ | $0.4418$ |

Table 5: Normalized CRPS scores on M5 and Favorita datasets. We average the deep learning based methods over 10 independent runs. The rest of the methods had very little variance. We report the corresponding standard error and only bold the numbers that are significantly better than the rest. The second best numbers in each column are italicized. We also report the mean performance across all levels in the corresponding column. Some of the baselines returned invalid quantiles (NaNs) and were omitted from the table.

| M5 | L0 | L1 | L2 | L3 | Mean |
|---|---|---|---|---|---|
| TDProb | *0.0227* ± 0.0001 | **0.0273** ± 0.0004 | **0.0304** ± 0.0004 | **0.1992** ± 0.0006 | **0.0699** ± 0.0002 |
| Prophet + historical fractions | 0.0227 ± 0.0 | 0.0289 ± 0.0 | 0.0409 ± 0.0 | 0.2612 ± 0.0 | 0.0884 ± 0.0 |
| Hier-E2E | 0.1143 ± 0.0039 | 0.1109 ± 0.0039 | 0.1175 ± 0.0039 | 0.2862 ± 0.003 | 0.1572 ± 0.0035 |
| DeepAR-BU | 0.0421 ± 0.0027 | 0.0442 ± 0.0023 | 0.0497 ± 0.0020 | *0.2092* ± 0.0003 | 0.0863 ± 0.0017 |
| DeepSSM-BU | 0.0294 | 0.0331 | 0.0420 | 0.2898 | 0.0986 |
| PERMBU-MINT | **0.0224** | *0.0281* | *0.0316* | 0.2147 | *0.0742* |
| ETS-BU | 0.0386 | 0.0490 | 0.0536 | 0.2905 | 0.1079 |
| ETS-MINT-OLS | 0.0356 | 0.0457 | 0.0508 | 0.2853 | 0.1043 |
| ETS-MINT-SHR | 0.0408 | 0.0498 | 0.0539 | 0.2856 | 0.1075 |
| ETS-ERM | 0.3491 | 0.3502 | 0.3676 | 0.9406 | 0.5019 |
| ARIMA-BU | 0.1116 | 0.1127 | 0.1162 | 0.3006 | 0.1602 |
| ARIMA-MINT-SHR | 0.0671 | 0.0729 | 0.0752 | 0.2896 | 0.1262 |
| ARIMA-ERM | 0.0590 | 0.0616 | 0.0731 | 0.4043 | 0.1495 |

| Favorita | L0 | L1 | L2 | L3 | Mean |
|---|---|---|---|---|---|
| TDProb | **0.031** ± 0.0005 | **0.0503** ± 0.0005 | **0.0711** ± 0.0006 | **0.1478** ± 0.0009 | **0.0751** ± 0.0005 |
| Prophet + historical fractions | **0.031** ± 0.0 | *0.061* ± 0.0 | *0.103* ± 0.0 | 0.224 ± 0.0 | *0.104* ± 0.0 |
| Hier-E2E | *0.0635* ± 0.0027 | 0.0944 ± 0.0023 | 0.1427 ± 0.0021 | 0.274 ± 0.0021 | 0.1437 ± 0.002 |
| DeepAR-BU | 0.0854 ± 0.0068 | 0.0920 ± 0.0062 | *0.1018* ± 0.0057 | 0.1744 ± 0.0054 | 0.1134 ± 0.0059 |
| DeepSSM-BU | 0.0979 | 0.1046 | 0.1372 | 0.3602 | 0.1750 |
| ETS-BU | 0.0987 | 0.1061 | 0.1206 | *0.1502* | 0.1189 |
| ETS-MINT-OLS | 0.106 | 0.1135 | 0.1355 | 0.1669 | 0.1305 |
| ARIMA-BU | 0.0856 | *0.0962* | 0.1205 | 0.2199 | 0.1306 |
| ARIMA-MINT-OLS | 0.1266 | 0.1444 | 0.1504 | 0.2084 | 0.1574 |

## B  Datasets

We use publicly available benchmark datasets for our experiments.

1. M5 [5]: It consists of time series data of product sales from 10 Walmart stores in three US states. The data consists of two different hierarchies: the product hierarchy and store location hierarchy. For simplicity, in our experiments we use only the product hierarchy consisting of 3k nodes and 1.8k time steps. Time steps 1907 to 1913 constitute a test window of length 7. Time steps 1 to 1906 are used for training and validation.

2. Favorita [6]: It is a similar dataset, consisting of time series data from Corporación Favorita, a South-American grocery store chain. As above, we use the product hierarchy, consisting of 4.5k nodes and 1.7k time steps. Time steps 1681 to 1687 constitute a test window of length 7. Time steps 1 to 1686 are used for training and validation.

3. Australian Tourism dataset[7]: consists of monthly domestic tourist count data in Australia across 7 states which are sub-divided into regions, sub-regions, and visit-type. The data consists of around 500 nodes and 228 time steps. This dataset consists of two hierarchies (Geo and Trav) as also followed in (Rangapuram et al., 2021). Time steps 1 to 221 are used for training and validation. The test metrics are computed on steps 222 to 228.

4. Traffic (Cuturi, 2011): Consists of car occupancy data from freeways in the Bay Area, California, USA. The data is aggregated in the same way as (Ben Taieb & Koo, 2019), to create a hierarchy consisting of 207 nodes spanning 366 days. Time steps 1 to 359 are used for training and validation. The remaining 7 time steps are used for testing.

5. Labour: Australian employement data consisting of 514 time steps sampled monthly, and 57 node hierarchy.

---

[5] https://www.kaggle.com/c/m5-forecasting-accuracy/

[6] https://www.kaggle.com/c/favorita-grocery-sales-forecasting/

[7] https://robjhyndman.com/publications/mint/

Table 6: We report point forecasting metrics on the Favorita dataset. We compare our model w.r.t the best baseline in terms of WAPE/ NRMSE (normalized RMSE) calculated using the p50 predictions of both the models.

| Favorita | L0 | L1 | L2 | L3 | Mean |
|---|---|---|---|---|---|
| TDProb | **0.0411 / 0.0523** | **0.0754 / 0.1829** | **0.1080 / 0.2639** | **0.2017 / 0.5983** | **0.1066 / 0.2743** |
| Best of rest (DeepAR-BU) | 0.0842 / 0.1065 | 0.0958 / 0.2644 | 0.1164 / 0.3194 | 0.21 / 0.6377 | 0.1266 / 0.332 |

Table 7: We report results on a longer horizon task on the Favorita dataset i.e where the test and validation size is $\tau = 35$. We compare our model with the best baseline for this dataset.

| Favorita | L0 | L1 | L2 | L3 | Mean |
|---|---|---|---|---|---|
| TDProb | **0.0428** $\pm$ 0.0004 | **0.0678** $\pm$ 0.0008 | **0.1085** $\pm$ 0.001 | **0.2503** $\pm$ 0.0022 | **0.1173** $\pm$ 0.0009 |
| Best of rest (DeepAR-BU) | 0.1551 $\pm$ 0.1065 | 0.1579 $\pm$ 0.02635 | 0.1645 $\pm$ 0.0225 | **0.2519** $\pm$ 0.0127 | 0.1823 $\pm$ 0.0224 |

6. Wiki2: This dataset is derived from a larger dataset consisting of daily views of 145k Wikipedia articles. We use a smaller version of the dataset introduced by Ben Taieb & Koo (2019) which consists of a subset of 150 bottom level time series, and 199 total time series.

For both M5 and Favorita we used time features corresponding to each day including day of the week and month of the year. We also used holiday features, in particular the distance to holidays passed through a squared exponential kernel. In addition, for M5 we used features related to SNAP discounts, and features related to oil prices for Favorita. For Tourism, Traffic, Labour, and Wiki2 we only used date features such as day of the week, month of the year, and holiday features from the GluonTS package (Alexandrov et al., 2020). All the input features were normalized to -0.5 to 0.5.

## C    Root Probabilistic Model

In this section, we provide more details about the implementation of our root probabilistic model using Prophet Taylor & Letham (2018). In the Prophet models we combined three components implemented by Prophet: local linear trend with automatic change point detection, linear regression for holiday effect, and Fourier series for seasonality effect. For M5 and Favorita datasets, we used weekly, monthly and yearly seasonality. For Tourism and Labor, we used monthly and yearly seasonality, for Traffic and Wiki we used weekly seasonality. We tuned hyperparameters *seasonality prior scale*, *holidays prior scale*, *changepoint prior scale*, *changepoint range* and *Fourier orders* using the validation set.

For M5 and Favorita datasets, we combined three models implemented by Prophet: local linear trend with automatic change point detection, linear regression for holiday effect, and Fourier series for seasonality effect. The seasonality model include weekly seasonality, monthly seasonality, and yearly seasonality. In the Prophet model for Australian Tourism dataset, we combined local linear trend with automatic change point detection and Fourier series for seasonality effect, where the seasonality effect is on "month of year". We tuned hyperparameters *seasonality prior scale*, *holidays prior scale*, *changepoint prior scale*, *changepoint range*, and *Fourier orders* using the validation set.

In order to show case that our model can be coupled with any root level model we also present results using the other models on L0. In the wiki dataset, we notice that the Deep-SSM (Rangapuram et al., 2018) works exceedingly well on the top-level. Therefore we also include a version of results where we combine our fractions model with the top-level forecast of the Deep-SSM model in Table 4. We can see that this `TDProbDeepSSM` model performs the best in terms of the mean CRPS across all levels. We also add results that combines our top-down model with an ARIMA model on L0. This `TDProbAR` model also ranks among the top three models (pretty close to the other two TDProb models) thus showing that we can use many different models on L0 and still achieve SOTA performance using our top-down proportions model.

Table 8: We report results on a longer horizon task on the M5 dataset i.e where the test and validation size is $\tau = 35$. We compare our model with the best baseline for this dataset.

| M5 | L0 | L1 | L2 | L3 | Mean |
|---|---|---|---|---|---|
| TDProb | **0.03** ± 0.0006 | **0.0376** ± 0.0002 | **0.0438** ± 0.0009 | 0.2882 ± 0.0021 | **0.0999** ± 0.0006 |
| Best of rest (PERMBU-MinT) | 0.0476 ± 0.0 | 0.0518 ± 0.0 | 0.0579 ± 0.0 | **0.2475** ± 0.0 | **0.1012** ± 0.0 |

## D  Full Results on All Datasets

Tables 4 and 5 show the full set of results for all datasets, baselines, and all hierarchical levels.

## E  Additional Experimental Details

**Hyper-parameters and validation.** The hyper-parameters used in our proportions models are learning rate (log scale 1e-5 to 0.1), number of attention layers ([1, 2, 4, 6]), number of attention heads ([1, 2, 4, 6]), LSTM hidden size ([16, 32, 48, 64]), batch-size ([4, 8, 32, 64]), output hidden layer after LSTM decoder (ff-dim) ([32, 64, 256]), node embedding dimension ([4, 8, 16]). We tune these hyperparams on validation loss.

The best hyperparameters for the different datasets are:

*Favorita.* learning-rate: 0.00085, fixed-lstm-hidden: 48, num-attention-heads: 16, num-attention-layers: 2, ff-dim: 16, node-emb-dim: 4, batch-size: 32

*M5.* learning-rate: 0.00079, fixed-lstm-hidden: 48, num-attention-heads: 4, num-attention-layers: 6, ff-dim: 64, node-emb-dim: 8, batch-size: 4

*Tourism.* learning-rate: 0.00031, fixed-lstm-hidden: 32, num-attention-heads: 8, num-attention-layers: 4, ff-dim: 32, node-emb-dim: 16, batch-size: 48

*Labour.* learning-rate: 0.003, fixed-lstm-hidden: 48, num-attention-heads: 16, num-attention-layers: 4, ff-dim: 64, node-emb-dim: 4, batch-size: 4

*Traffic.* learning-rate: 0.003, fixed-lstm-hidden: 48, num-attention-heads: 16, num-attention-layers: 4, ff-dim: 32, node-emb-dim: 4, batch-size: 4

*Wiki.* learning-rate: 0.000295, fixed-lstm-hidden: 48, num-attention-heads: 16, num-attention-layers: 4, ff-dim: 64, node-emb-dim: 4, batch-size: 4

**Training details.** Our model is implemented in Tensorflow (Abadi et al., 2016) and trained using the Adam optimizer with default parameters. We set a step-wise learning rate schedule that decays by a factor of 0.5 a total of 8 times over the schedule. The max. training epoch is set to be 50 while we early stop with a patience of 10. All our experiments were performed on a single server with a 32 core Intel Xeon CPU and an Tesla V100 GPU.

**Baselines.** We used the experimental framework released by Rangapuram et al. (2021) for running the baselines in the former paper. For the rest of baselines (`DeepAR-BU` and `DeepSSM-BU`) we used GluonTS (Alexandrov et al., 2020) implementations of Deep-SSM and DeepAR.

