# OpenReview forum: "A Deep Top-Down Approach to Hierarchically Coherent Probabilistic Forecasting"
_TMLR — Rejected by TMLR_

### Review · Reviewer_u2H7 · 2022-12-16

**Summary Of Contributions:**

The paper considers the problem of probabilistic hierarchical forecasting where the goal is to generate coherent probabilistic predictions for a large number of time series arranged in a hierarchical structure. Specifically, a new top-down approach to hierarchical forecasting is proposed. An attention-based RNN model is used to learn the distribution of the proportions, which are then coupled with an independent univariate probabilistic forecasting model for the root time series. Experiments on several public datasets show signiﬁcant improvements on most datasets compared to state-of-the-art methods. The authors also provide a theoretical justiﬁcation of the top-down approach compared to bottom-up methods (in an idealized setting).

**Audience:**

No

**Broader Impact Concerns:**

NA.

**Claims And Evidence:**

No

**Requested Changes:**

Major comments:

- Experiments
	- It is not clear if the comparison between methods is fair. For example, how did you generate probabilistic forecasts from methods than only generate point forecasts?
	- "Note: Some of the baselines listed in Sec. 5 returned invalid quantiles (NaNs) and were omitted from the table."
		- Removing methods because they returned NaNs is not the best approach when comparing baselines. Why did they return NaNs? Also, is that the only reason why some methods do not appear for some datasets?

- The authors talk about "coherent probabilistic forecasts", but do not define this concept. The coherency paragraph in Section 2 is not clear. Citations are needed there. Different definitions of coherency have been proposed in the literature. What is the difference between $\hat Y$ and $\hat Y_F$? Also, Y_F is an F x N matrix, how is coherency defined for a matrix?


- The "related work" should clearly separate the literature on point and probabilistic forecasting. Or at least clearly state which method can generate "coherent" probabilistic forecasts.

- "Furthermore, none of these approaches can directly handle probabilistic forecasts."
	- I disagree with this statement. The MinT approach provides coherent means and a covariance matrix for the entire hierarchy, which enables probabilistic forecasting.

- "Their approach did not complete training within 2 weeks": what kind of computer architecture did you use? Running a forecasting algorithm on such "small" datasets for two weeks seems inappropriate.

Other comments:
- In the introduction, it is stated that the parameters are obtained using an RNN, but later, the authors talk about an LSTM.
- What is p50?
- Section 4 and appendix:$i$ goes from $t$ to $n$, then $i$ goes from $t$ to $n$, and in the appendix, $i$ goes from $1$ to $n$.
- Did you average the forecast error over the entire forecast horizon? This should be discussed in the text.
- In expression (3), you use $Q_s^i$ but you only defined $Q_F^i$.
- What do you mean by "predictive distribution quantiles"?
- Theta_i^b = p_i = theta_hat. Did you mean theta_0^t?
- Did you mean x_t (instead of x_i) in Section 4?
- Typos: "each of row", "data distribution data", "an results"
- "Note that we only need to load all the time-series of a given family into a batch.": Why is that the case?
- y_hat and b_hat are not defined

**Strengths And Weaknesses:**

Strength
- Probabilistic forecasting for hierarchical time series is a challenging and important research problem for many applications.
- The proposed method can be used with any forecasting method for the root time series and generates coherent probabilistic forecasts by construction.
- The authors compare their method with multiple baselines and datasets.
- A theoretical analysis is provided to justify the advantage of a top-down approach over a bottom-up method.


Weaknesses

- The paper is poorly written (see Requested Changes). The terminology is not properly introduced, e.g. "probabilistic, hierarchically-coherent models", coherent probabilistic forecasts", "revised coherent forecasts", "the samples are reconciled", "root time series", "hierarchy tree", "leaf forecasts", "child", "same family", etc.

- It is not clear how the theoretical analysis relates to the proposed method. It seems that it motivates the top-down method over a bottom-up approach in an idealized setting, which is not new. Also, the authors should clearly discuss the assumptions and limitations of their analysis. For example, the analysis is done for a method based on historical proportions.

- Experiments
	- It is not clear if the comparison between methods is fair. For example, how did you generate probabilistic forecasts from methods than only generate point forecasts?
	- "Note: Some of the baselines listed in Sec. 5 returned invalid quantiles (NaNs) and were omitted from the table."
		- Removing methods because they returned NaNs is not the best approach when comparing baselines. Why did they return NaNs? Also, is that the only reason why some methods do not appear for some datasets?

- Except for a "small" improvement in forecast accuracy, the methodological contribution of the paper is weak. While the paper proposes a new top-down method, it is not clear how it advances the state-of-the-art given that there are other end-to-end probabilistic forecasting methods that are as general as the proposed method.

---

> ### Author Response · Authors · 2022-12-22
> **Response - Part 1**
>
> Thank you for the valuable feedback. We have addressed your concerns below. We have already updated them in the paper however we will post the revised paper after all three reviews are submitted  (as recommended in the official guidelines: https://jmlr.org/tmlr/editorial-policies.html)
>
> - Terminology: We will update the paper to introduce the terminology in a better way in the next version of the paper. In particular we will clarify the meaning of the following terms/phrases:
>
>   "probabilistic, hierarchically-coherent models", coherent probabilistic forecasts": probabilistic forecasts such that samples from the predicted distributions are coherent. This definition has been added in Section 2.
>
>   "revised coherent forecasts", "the samples are reconciled": post-hoc reconciliation of base forecasts that are not coherent. We have added matrix definitions along with citations.
>
>   "root time series": the time-series at the root of the tree that is the sum of all time-series in the dataset.
>
>   "leaf forecasts": forecasts for time-series at the leaf of the tree (finest granularity)
>
>   "Family": an internal node in the tree along with all its children.
>
> - Theoretical analysis: In this paper, we demonstrate the advantages of a top-down approach theoretically, under certain assumptions as discussed in Section 4. In the revised version of the paper, we clearly emphasize the limitations of this analysis. In particular, our theoretical justification only applies to learning historical proportions; it would be interesting to extend it to predicted future proportions.
>
>   We are not familiar with any other work that performs a theoretical analysis comparing top-down and bottom-up, even in an idealized setting. We would be happy to discuss and compare ours with any existing approach, if a citation to the work is provided. In Section footnote 3 we also differentiate why our method circumvents the independence assumption of base forecasts and therefore need not be biased according to Hyndman et al 2011.
>
> - Baselines: The official implementation (Hyndman et al., 2015) of the baselines PERMBU-MINT, ETS-MINT-SHR, ETS-ERM, ARIMA-MINT-SHR, ARIMA-MINT-OLS, and ARIMA-ERM returned invalid values (possibly because of numerical issues) either for the Favorita or the M5 dataset - our largest datasets, and were omitted from the table. These baselines do succeed for every other dataset.
>
>   We also used the public implementation of SHARQ, and it trains an independent model for each node. The favorita dataset for example has 300+ internal nodes. Training such a model on this dataset on a similar architecture (V100 GPUs) does not finish training even after a few weeks, as compared to the other approaches which don’t take more than a few hours per run.
>
> - Comparing probabilistic and non-probabilistic predictions:  For the methods that produce point forecasts, we use the point predictions for calculating the quantile losses for all quantiles in the CRPS expression. This is the same convention that was followed in the prior work (Rangapuram et al., 2021). We have now clearly separated the point vs probabilistic methods in the tables for better readability.
>
> - Significance of improvement: As noted in the paper, we show improvements of up to 27% compared to previous baselines, which is significant. For instance this is more than the improvements of up to 19% shown by Rangapuram et al., 2021 over earlier baselines, even though we use a superset of the datasets and baselines in that paper. Furthermore, we show an average (across all datasets) improvement of 18% over Rangapuram et al., 2021, which was the previous state-of-the-art work in this space . We would be happy to address any further concerns about this.
>
> - Related work: Thanks for the helpful suggestion! We have updated the related works section so it is clearer.
>
> - Probabilistic forecasts from MinT: Thanks for pointing this out. We will add a discussion in the next version of the paper that MinT forecasts can be converted into probabilistic forecasts with a Gaussian assumption, by using the empirical covariance matrix. PERMBU-MinT is an improvement over that and we compare it directly.
>
> (continued in part 2)

---

> > ### Comment · Reviewer_u2H7 · 2023-01-19
> > **More comments and where is part 2?**
> >
> > We thank the authors for their response. However, we do not see the "part 2" response.
> >
> > Here are additional comments:
> >
> > - "returned invalid values (possibly because of numerical issues)"
> >
> > Explaining NaN values for state-of-the-art methods by speculating numerical issues does not feel appropriate to me, especially for a paper that has very limited methodological contributions. This suggests that the authors did not even try to understand the SOTA methods.
> >
> > Furthermore, even if the official implementation fails (which I doubt), there are other implementations. See https://github.com/Nixtla/hierarchicalforecast
> >
> > - "Training such a model on this dataset on a similar architecture (V100 GPUs) does not finish training even after a few weeks"
> >
> > From the SHARQ paper, we can read "SHARQ is time and memory efficient, scaling well in both aspects with large datasets. One can simultaneously train multiple time series and keep a running sum for reconciliation". Also, experiments in the paper have considered hierarchies with 755 time series and 42840 time series. You have issues with a hierarchy containing 300+ series, which is abnormal.
> >
> > - What do you mean by "point predictions"? Which predictions? mean? median?
> >
> > - To me, it does not make sense to have a paper on probabilistic hierarchical forecasting that does not include comparisons with the SOTA MINT probabilistic forecasts.

---

> > > ### Author Response · Authors · 2023-01-24
> > > **Response**
> > >
> > > We apologize for part 2 of our response not being visible. We accidentally set the wrong visibility. It should be available now.
> > >
> > > We thank the reviewer for the helpful suggestions.
> > >
> > > **Missing PERMBU-MinT from Favorita:** We thank the reviewer for pointing out the nixla package. We are running the PERMBU-MinT implementation in that package on our favorita dataset as of now. It has been running for more than 24 hrs without finishing, and we will post the results as soon as it finishes.
> > >
> > > Regarding the original implementation, we assure you that we run the correct version of the [code](https://github.com/rshyamsundar/gluonts-hierarchical-ICML-2021) that is a python wrapper around the original implementation released as a part of the HTS R package. In order to showcase this we have packaged the original package along with our dataset in the supplementary. We also include a README so that the reviewer can run the code and check the NaN results if they please. We are investigating further why this is the case but it is beyond our scope to investigate every detail of the R package (even though we do understand and appreciate the work in the original paper).
> > >
> > > **Regarding SHARQ:** Please note that the original implementation released by the authors is only set up to run on very small datasets (please see the [link](https://github.com/aaronhan223/htsf/tree/main/SHARQ)). We use the original package on our larger datasets and it fails to finish as stated. We have reached out to the authors to clarify the same. Meanwhile we have included the packaged code and our datasets with a README in the supplementary so that our observations can be reproduced.
> > >
> > > **Regarding point predictions:** We think there might be a misunderstanding. Methods like ETS-BU, ARIMA-BU, ETS-ERM etc are set up in their original implementation to only produce point forecasts i.e there is no question of mean or median as in probabilistic forecasts. These point forecasts are used for every quantile to calculate the CRPS score. We followed this strategy to be faithful to the comparisons in Hiere2e paper (they do the same thing for all non-probabilistic baselines).
> > >
> > > **MinT being SOTA:** PERMBU-MinT and Hiere2e are SOTA methods for coherent probabilistic forecasting. We do compare with Hiere2e on every dataset and PERMBU-MinT on every dataset other than Favorita (as discussed above).

---

> > > > ### Author Response · Authors · 2023-01-26
> > > > **An update on experiments**
> > > >
> > > > Following the reviewer's great suggestion we tried to work with the nixtla package on Favorita. Naively applying the PERMBU-MinT methods on the Favorita dataset fails due to two problems (it succeeds on all other datasets and numbers have been reported already in the paper):
> > > >
> > > > 1. The reverse engineered covariance calculation yields zero or negative covariance due to time-series whose predictions and values are all zeros in the test period. The root cause of this is in the logic of this [function](https://github.com/Nixtla/hierarchicalforecast/blob/e5d79fd2b97ce04e9943a1749915f3b9ad862ddd/hierarchicalforecast/core.py#L39).
> > > >
> > > > 2. The covariance of the residuals calculated [here](https://github.com/Nixtla/hierarchicalforecast/blob/e5d79fd2b97ce04e9943a1749915f3b9ad862ddd/hierarchicalforecast/methods.py#L636) is not positive definite and therefore MinT cannot be applied. This is again because of presence of all zero residuals.
> > > >
> > > > We fixed the first problem by making the "sigmah" a small positive value of 1e-5 for coordinates in which it was zero or negative. The second problem could not be fixed unfortunately. We tried to add small amts of noise which makes it positive definite but the results from MinT  have CRPS > 1.0 for some levels possibly due to numerical instability from small eigen-values of the covariance matrix. For example  one such run produced the scores:
> > > >
> > > >
> > > > | level   | CRPS        |
> > > > |---------|--------------|
> > > > | 0       | 0.1234+/-0.0 |
> > > > | 1       | 1.7654+/-0.0 |
> > > > | 2       | 2.7595+/-0.0 |
> > > > | 3       | 4.3942+/-0.0 |
> > > > | mean    | 2.2606+/-0.0 |
> > > >
> > > > Therefore we went ahead with the fix of (1) and PERMBU-Bottomup. This gives us a quite poor result overall:
> > > >
> > > >
> > > > | level   | CRPS         |
> > > > |---------|--------------|
> > > > | 0       | 0.078+/-0.0  |
> > > > | 1       | 0.8451+/-0.0 |
> > > > | 2       | 0.6455+/-0.0 |
> > > > | 3       | 0.2368+/-0.0 |
> > > > | mean    | 0.4514+/-0.0 |
> > > >
> > > >
> > > > In contrast just doing bottom up with the same base forecasting method (probabilistic AutoARIMA) gives much more reasonable values (still worse than our method):
> > > >
> > > > | level   | CRPS         |
> > > > |---------|--------------|
> > > > | all     | 0.1233+/-0.0 |
> > > > | 0       | 0.0775+/-0.0 |
> > > > | 1       | 0.0816+/-0.0 |
> > > > | 2       | 0.0967+/-0.0 |
> > > > | 3       | 0.2373+/-0.0 |
> > > > | mean    | 0.1233+/-0.0 |
> > > >
> > > > We can include the PERMBU-BU numbers and the discussion above if it would please the reviewer.

---

> > > > > ### Author Response · Authors · 2023-02-01
> > > > > **More updates**
> > > > >
> > > > > In order to make sure that the reported numbers from Nixtla hierarchical forecasting are indeed correct, we ran PERMBU-BottomUP with base forecasts from AutoARIMA on our m5 dataset and the numbers are as follows:
> > > > >
> > > > >
> > > > > | level   | CRPS     |
> > > > > |---------|--------------|
> > > > > | 0       | 0.102+/-0.0  |
> > > > > | 1       | 0.0993+/-0.0 |
> > > > > | 2       | 0.0984+/-0.0 |
> > > > > | 3       | 0.257+/-0.0  |
> > > > > | mean    | 0.1392+/-0.0 |
> > > > >
> > > > > The above numbers are quite reasonable though still worse than our model's numbers. Therefore the conclusion seems to be that PERMBU (even from the Nixtla package) does not form a good baseline on the Favorita dataset.

---

### Review · Reviewer_sRV3 · 2022-12-23

**Summary Of Contributions:**

The submission is about probabilistic forecasting of timeseries, where the latter decompose hierarchical following linear constraints. The proposed approach ensures that the decomposition (sum-aggregation) constraints are satisfied by design when generating the children (i.e. more fine grained) timeseries conditioned on the ancestor (starting from an initial root-level prediction). This is achieved by training a seq-to-seq model to output sample disaggregation coefficients which are elements of a vector on the simplex (size of the node outdegree).
The submission provides a theoretical result on a specific setting (2-level hierarchy) showing that the excess risk of a bottom up approach against the proposed model is bigger (in the worst case). An empirical analysis is provided on 6 popular benchmarks, showing evidence of a competitive advantage of the proposed model.

The key contribution in the submission  is seemingly a quite straightforward and efficient way of obtaining consistent hierarchical timeseries predictions. The theoretical result is a nice side addition, but limited in impact.


**Audience:**

Yes

**Broader Impact Concerns:**

I have no concerns on this matter.

**Claims And Evidence:**

No

**Requested Changes:**

-- MAJOR CHANGES (influencing paper acceptance)

* The discrepancies between the performance results in this paper and those in Rangapuram et al (2021) need to be either convincingly discussed or amended. Results in Table 1 for Labour/Traffic/Wiki/Tourism report a performance for SOTA models that is substantially lower than that in Rangapuram et al (2021). Since the same setting is being used (aside from repeating the experiment 10 times instead of 5), it is not clear to me how results could be so divergent (way outside of the intervals of the standard deviations reported in this submission and in the Rangapuram et al 2021 paper). Note that in paper [1], similar experiments are performed and the performance of HierE2E is in line with what reported in Rangapuram et al 2021, rather than to the values in Table 1 of this submission.

 * Why is PERMBU-MINT result only reported for M5? It should be made available for all datasets in Table 1. Note for instance that Rangapuram et al 2021 report results for PERMBU-MINT on Labour, Traffic, Tourism, Wiki.

 * Can you please elaborate on whether the approach can be used on multi-level problems? If so, it would be best to test it also on Tourism-L.

 * To me it is not entirely clear how the model is trained. This stems from the fact that I am not completely getting the formulation of the Dirichlet log-lik loss in (2) and how this is optimized (and the argument about TFProbability-does-it-all-for-us does not help). For instance, if I look at (2) I am missing the two $\Gamma(\alpha_i)$ sum-log and log-sum terms. Are they absorbed in the formulation $log(B(\alpha))$? If so, this is not helping and I suggest to write the log-lik explicitly. Another thing that is not clear to me is how the sufficient statistics $a_i$ are obtained? In a standard log-lik maximization I expect to be able to estimate those from supporting observed multinomial data. How are these $a_i$ obtained here?

 * Also using $\alpha_i$ for the Dirichlet concentration and $a_i$ for the proportions does not really help readability.

[1] https://arxiv.org/pdf/2110.13179.pdf

-- MINOR CHANGES

 * Please tune down some of the overstatements in the paper or provide convincing evidence for them to be well-founded

 * Consider tiding up the notation following the comments in this and the previous review report section.

* The submission could use a broadening and deepening of the discussion of related works. I know that these are very recent and preprint, but I suggest looking into the following papers which seem highly related:
https://arxiv.org/pdf/2206.07940.pdf
https://arxiv.org/pdf/2110.13179.pdf


**Strengths And Weaknesses:**

-- Strenghts --

(+) The paper follows the structure, ideas, concepts and benchmarks of a tight community within the ML field (that of hierarchical forecasting). This community is well situated within the TMLR readership and the submission itself does touch on the points that are expected by typical works from this community. There is also a minor issue in this, as the work is not really ground-breaking when it comes to providing insights and a novel perspective over the problem.

(+) The intuition underlying the approach is quite clear and solid: generating the disaggregation coefficients by learning a conditional Dirichlet-like distribution through a seq-to-seq model seems efficient and effective. Overall this intuition seems also novel with respect to the literature on hierarchical  timeseries forecasting.

(+) The empirical analysis is quite broad and the results, if one limits the reading to the numbers in the paper, shows that the approach has a substantial edge over the state of the art models.

-- Weaknesses --

(-) The paper has a certain tendency to overstate, e.g.:
 * In Section 2, it is stated that PERMBU is the only reconciliation-based hierarchical approach for probabilistic forecast. This does not seem true: Panagiotelis et al (2020), at least, also deal with the same problem. A more careful and precise discussion of the related works seems needed, as this is a journal publication, not a conference paper.
 * In Section 5, it is stated that the dataset used in this work are a strict superset of those used in other paper. This is not entirely true as  [1] (for instance) is assessed also on Tourism-L, Flu-Sympthoms. FB-survey, which are not present in this study.

[1] https://arxiv.org/pdf/2110.13179.pdf

(-) Section 3 is quite confusing in presenting the model, with an incremental stacking of additional notation which is not always needed and which does not always help shedding light on the model. For instance, $t$ is typically used to index time in Section 2, but then in Section 3 also $s$ is used for the same purpose, without any clear explanation of the why. Further the decoder outputs $D_\mathcal{F}$ are provided as-is with no clear definition of how these should be interpreted and the role they play in the overall model.  Additionally, dimension $F$ pops-up when defining $B_{\mathcal{F}}$ but again no definition is provided for it (I guess it is the size of the future samples to be forecasted). My suggestion is to do a careful tidy-up of the notation and to make sure to define any new symbol/term that is introduced.

(-) Details about how the model is trained to generate the disaggregation coefficients are not provided. Much is left to the reader to figure out how the seq-to-seq model is trained to predict such proportions (more on this on the requested changes)

(-) The empirical results are not consistent with the numbers in Rangapuram et al 2021, despite the paper stating that the same empirical setting is used. Also, the selection of literature models changes with the datasets, making it hard to consistently position the performance of the model w.r.t. the state-of-the-art.

(-) No indication is provided on whether the approach can handle multi-level hierarchies. I suspect this is not the case as TOURISM-L (which is typically used to assess this aspect) is not present in the benchmarks.
(-) The initial part of the submission builds motivations heavily on the importance of having a probabilistic forecasting model to measure prediction uncertainty (it is presented as a major feat of the work). But this aspect is never assessed later on in the paper, nor the probabilistic formulation of the model is ever formalised/used later on (only very brief mention is that the seq-to-seq is trained to mimic a Dirichlet distribution).

---

> ### Author Response · Authors · 2023-01-09
> **Response**
>
> We thank the reviewer for their careful reading of the paper and the great suggestions. Below we address the concerns:
>
> **More accurate statements.**
>
> _About PERMBU-MinT being the only reconciliation based approach:_ We thank the reviewer for pointing this out. We have revised the language in this part and also added a citation to the mentioned paper.
>
> _About the baselines and dataset:_ In the original paper we had mentioned that the baselines are a strict superset not the datasets. We have made the statement even more precise in the revised paper which now states that the baselines are a strict superset of the ones used in the Hiere2e paper.
>
> **Notation in Section 3:** We apologize for the confusion. We have added more details to the general notation setup in Section 2. Note that all future data-points denoted by the subscript $\mathcal{F}$ have $F$ time-steps as clarified in the beginning of Section 2. We have now added a callback to this when we first introduce F in Section 3.
>
> $t$ is still used as the time-index in Section 3. We additionally use “s” when “t” has a special meaning of being the first index of the prediction time-period or when we sum over time $t$ to $t + F - 1$.
>
> **Dirichlet loss function and dis-aggregation notation:** We have added the full definition of the loss function to the revised paper and also clarified why we can backpropagate through the loss functions. The $a$ in the loss function are not sufficient statistics but are the true proportions of the children nodes in that family in the future that the model is trying to learn. During training they are simply obtained by dividing the child node’s datapoint by the parent node’s datapoint at a particular time-step. This has been clarified in Equation 1 and we have added a callback reference to this in the revised version of the paper.
>
> We have also changed the notation of the Dirichilet parameters to $\pmb{\beta}$ to make it more readable.
>
> **Discrepancies w.r.t to HierE2E:** We apologize for the confusion. We use identical settings in Tourism-L and Labour datasets w.r.t to the HierE2E paper and the numbers reported are also identical. However, for traffic and wiki2 datasets we noticed (during the original submission) that the test and validation size was only one time-step in the Hiere2e setup which is incredibly small. Moreover on the traffic dataset the test period is the day of Dec 31st with _less than a year of data_. So none of the models have any chance of learning the fact that Dec 31st is an atypical time-point in terms of traffic. Therefore we extend the test and validation period to 7 days for a more robust test setup. We also use the original code provided by the authors of HierE2E  for model training and evaluations of the baselines. This is evident from the fact that HierE2E does indeed show strong performance on these datasets as in the original paper.
>
> This information was in the appendix in the original submission. We have now added back the table describing the dataset settings to the main paper. We have also added the above explanation to Section 5 of the revised paper.
>
> **PERMBU-MinT comparison:** PERMBU-MINT is reported for every dataset except for Favorita (the largest dataset) because on that dataset the original implementation produced NaN values. For the sake of space in Table 2 of the main paper, we report the deep learning models separately and the “best of the rest” of the other models. This “best of the rest” includes PERMBU-MINT. We also provide all the numbers across all levels in Table 4 in the appendix. Even there “best of the rest” includes the best among all other baselines including PERMBU-MINT.
>
> **Tourism-L dataset:**  The approach can be used on Tourism-L and we report numbers of Tourism-L in the paper. The “Tourism” dataset in our table is actually Tourism-L and we specify this in the “Datasets” paragraph in Section 5 of the paper.
>
> We assume that the reviewer means multiple hierarchy and not multiple levels (as all our datasets have multiple levels). To apply our method to the Toursim-L dataset we calculate proportions for both the Geo and Trav hierarchy.

---

> > ### Comment · Reviewer_sRV3 · 2023-01-20
> > **Post-Rebuttal**
> >
> > I appreciate the clarifications provided by the Authors: the formal description of the approach is much clearer to me now. Also a more precise statemement of the empirical setup and of the usage of the benchmarks helps very much replicability and fairness of comparison with the state of the art.
> > However, while I understand that the straight use of original code for some of the methods might not have provided sensible results, I would have expected a stronger effort to provide such results through alternative means as other works in literature (referenced both by this submission and by the review) clearly succeded in running of the same methods on the datasets that are used in this manuscript.
> > Alternatively, some deeper discussion and analyses are espected other than a general reference to possible numerical methods.

---

> > > ### Author Response · Authors · 2023-01-24
> > > **Further response.**
> > >
> > > We thank the reviewer for the helpful  comments. It is true that one of the papers that was brought to our attention during this review process (https://arxiv.org/pdf/2110.13179.pdf) does report the PERMBU-MinT numbers on Favorita.
> > > We did run the original PERMBU-MinT codebase on the Favorita dataset and it produced NaN values (even though it ran successfully on all our other datasets, as we reported in the paper). We have packaged the code and the dataset in the supplementary material along with a README so that this issue can be verified if needed . Meanwhile we are also running the PERMBU-MinT implementation in the nixla package pointed out by Reviewer u2H7. However, even the code in the nixla package has been running for more than 24 hours on Favorita at this point, without finishing. We will report the results as soon as it finishes successfully, and will add them to the final version. Please note that we already report PERMBU-MinT numbers on all other datasets except Favorita.

---

> > > > ### Author Response · Authors · 2023-01-26
> > > > **An update on experiment**
> > > >
> > > > We have provided a detailed update on running PERMBU-MinT on the Favorita dataset to reviewer u2H7. Please have a look at that response and let us know if you have any further questions.

---

### Review · Reviewer_cNLu · 2022-12-27

**Summary Of Contributions:**

This paper proposes a probabilistic top-down approach for multivariate time series forecasting where the variables have a tree topology (in contrast to the more popular bottom-up approach).

The main contribution is predicting the future proportions of the children for a parent node and this method can be combined with any off-the-shelf univariate time-series modeling technique to generate a forecast for the entire tree hierarchy.

The authors use a seq2seq model (LSTM) shared across the children nodes to encode historical information followed by Transformer-like self-attention blocks to model the interaction between the children by predicting the parameters of a Dirichlet distribution. The model is optimized using the Dirichlet log-likelihood loss and cheap inference can be carried out by sampling from this learned distribution.

The authors also prove the lower bound for improvement compared to a bottom-up approach in a simplified setting.

**Audience:**

Yes

**Claims And Evidence:**

Yes

**Requested Changes:**

- I would like to see forecast results over different forecast windows
- I would be curious to see how the model performs with positional embeddings rather than learned embeddings.

**Strengths And Weaknesses:**

Pros:
- The paper is really well-written and easy to follow.
- The idea is simple and neat.
- The paper backs up all the claims with experiments and theoretical results.
- The theoretical justification for the top-down setting is fairly convincing (for the simplified setting)
- The appendix is quite thorough

Concerns:
- I would like to see forecast results over different forecast windows. Currently, only a fixed forecast window is used.
- Is choosing a single family in a mini-batch to optimize over the best approach? This would mean that the predictions only use local topological information and discard any other global structural information if the tree has more than 2 layers.
- Is a unique learned embedding required for all the nodes? Unlike a vocabulary look-up table, the tree topology has a structure that can be used to generate positional embeddings instead of learning it from scratch. Apart from reducing the optimizable parameter count, it could help with generalization as the model knows which tree level it is operating at.

Other Remarks:
- I found the characterization of the datasets missing in the paper only to find it in the appendix. It would be nice if the authors can mention in the paper that the dataset is characterized in the appendix.
- I worked through the theorems in the paper and they seem correct however it is possible that I might have missed something

---

> ### Author Response · Authors · 2023-01-09
> **Response**
>
> We thank the reviewer for their careful reading and the great suggestions.
>
> **Different Forecast Windows:** We thank the reviewer for this great suggestion. On datasets other than M5 and Favorita we followed the convention in Rangapuram et al 2021 for the sake of reproducibility. In Table 7 in the appendix we had already provided a different window result for the Favorita dataset where the task is to predict for the next $\tau=35$ days. We see a gain of 35% over the best baseline in this task in terms of mean CRPS score across all levels.
>
> In the limited time available during the rebuttal, we evaluated on a similar task on the m5 dataset where $\tau=35$. The results have been added to Table 8 in the revised paper. We see that as before numbers of our model are bold in all levels except L3 (where PERMBU-MinT does better). We can add longer horizon results for the other datasets in the paper as well.
>
> **One family in a mini-batch:** This is a great question. Requiring to load only one family in a mini-batch is actually an advantage of our method in terms of scalability. In industrial scale datasets it would be prohibitive to load all the time-series in the dataset into the GPU memory in one batch, which is required by prior works like HierE2E (Rangapuram et al 2021). We empirically show that even though we do not load all the time-series in a mini-batch our methods can outperform other methods that do. It would be interesting to look at methods to incorporate more global information in future work.
>
> **More structured node embeddings:** This is a great suggestion for future work in this area. We are assuming that the reviewer is talking about using tree based positional embedding like the ones in [this paper](https://papers.nips.cc/paper/2019/file/6e0917469214d8fbd8c517dcdc6b8dcf-Paper.pdf). This sounds like a great idea but it is slightly orthogonal to the main ideas in the paper and therefore beyond the scope of this paper.
>
> **Other remarks:** We have added the dataset table to the main body of the paper and have added more text description as well.

---

### Review · Reviewer_N1br · 2023-01-10

**Summary Of Contributions:**

This paper presents a top-down approach of hierarchical regression by estimating the proportions of dependent variables, not the value itself.

**Audience:**

Yes

**Broader Impact Concerns:**

Could be usable, but I need the weakness to be resolved, first.

**Claims And Evidence:**

Yes

**Requested Changes:**

It will be better to have a graphical illustration on constructing a matrix R matching the hierarchy in further details. Figure 1 (Right) shows a tree and a matrix, but the tree is too simple with only a two level. What if we have three level tree with 1 root, 3 child nodes, (3,4,2) grand child nodes? How to represent that in R? Also, place the figure on the same page of Section 2. Please do not cluster figures and put it on a distant page.


**Strengths And Weaknesses:**

Strength:

In terms of the structure and loss function design, I agree with the authors. LSTM would be a natural approach to the time-series estimation whether it is either proportion or time-series. If you are estimating the proportion, the Dirichlet distribution could be a good distribution of the proportion's sampling distribution.


Weakness:

1.
Given the definition on the hierarchy on Section 2, authors do not assume the case of assigning a dependent variable Y to the mid-level of the hierarchy. Is that right? In that case, the coherency is simply achieved best if all leaf-level Y are being perfectly fitted, is that also right? If both are right, the coherency will act as a regularizer, fundamentally limiting the fitness of Y, if we assume an optimal estimator for individual Y.

2.
This paper provides a dynamic estimation on the hierarchical proportions of aggregating Y. This dynamic estimation hinges upon that the proportion estimation is easier than the individual time-series estimation, which is claimed by authors. Is this right? and Why is it easier? That should actually be asked in the paper, and reasons should be provided.

From my perspective, the proportions are basically a simple statistics if you have a correct time-series estimation. Therefore, the proportions and the time-series Y should have similar difficulties in its estimations. What am I missing?

3.
In this aspect, the value of Theorem 4.1. is also questionable because its value depends upon a good estimation of $a_t$. The good estimated $a_t$ means that we know good information on each individual time-series, so the bottom-up approach is being a good estimator on Y, already.

---

> ### Author Response · Authors · 2023-01-12
> **Response**
>
> We thank the reviewer for the insightful questions and suggestions. Below we address the main concerns:
>
> 1. We assume that the reviewer means that predicting a level individually would produce the most accurate results at that level (please let us know if this is not an accurate interpretation of the question). This is a great question which is at the heart of coherent forecasting and we would like to provide a couple of points arguing why this is not always the best thing to do:
>
> - Firstly, in many applications coherency is a requirement, not a choice. For instance in the context of forecasting the demand of different products on a retail hierarchical tree if the forecasted samples are incoherent the forecasts might be hard to interpret and cause issues with down-stream consumers of the forecasts such as warehouse inventory planning systems. There have been numerous works on coherent forecasting precisely because in many cases this is a hard constraint (see Hyndman et al 2011).
>
> - Secondly, in some datasets, as the reviewer correctly points out, coherency can act as a good regularizer  and improve accuracy across several levels of the dataset. This is especially the case where leaf time-series are sparse, noisy and difficult to learn. This has been observed in prior works (i) Rangapuram et al 2021 note this in Section 3 (“Accuracy-coherence trade-off) (ii) Figure 2 in (Wickramasuriya et al., 2019)
>
>
> 2. The main intuition behind the success of our method is that there can be several scenarios where the fractions are more easier to predict than the root time-series:
>
> - Consider a retail hierarchical forecasting setting where the leaf levels' time series are the demand of different sizes for a specific item (e.g a particular brand of shoes or shirt); the parent time-series being the total sales for that item across all sizes. In this case the proportion of the population in different sizes either stays constant or changes very slowly with time. Therefore in this case the fractions might be easier to learn than just fitting all the time-series separately.
>
> - The fractions are also scale free i.e they are bounded between [0, 1]. In deep times-series forecasting [literature](https://openreview.net/forum?id=cGDAkQo1C0p) it has been observed that scale variations between different time-series can impact model performance negatively because of optimization issues and therefore need special considerations. Learning the fractions as opposed to the unnormalized values can also help with this issue.
>
> 3. The value of Theorem 4.1 is showing that in the idealized setting we considered, top-down approach is provably better than bottom-up approach. A good estimate of $a_t$’s does not necessarily mean we have good information on each individual child's regression coefficients, as we discussed in the retail example above. In fact, Theorem 4.1 holds since estimating $a_t$ is easier than estimating each individual child’s regression estimate.
>
> __Requested Changes:__ We thank the reviewer for this suggestion. We have added the said figure with the corresponding matrix in Figure 1 of the paper which is now embedded near the text of Section 2.

---

### Decision · Action_Editors · 2023-02-02

**Recommendation:** Reject

**Comment:**

The paper presents several weaknesses:
- presentation should be improved;
- contribution is incremental/derivative;
- there are no theoretical contributions to support the main claim of the paper;
- empirical assessment is not solid enough, due to incomplete comparison versus SOTA approaches (both in terms of proper execution of some of them, and in missing approaches, such as MinT) and in considered benchmarks (only one considered and modified from its original definition).

**Audience:**

Given the above reported weaknesses, I doubt that there could be individuals in TMLR'a audience interested in the paper. In addition, the proposed idea is incremental/derivative.

**Claims And Evidence:**

The paper claims are not supported by theoretical findings. Moreover, the experimental evidence provided by the authors is questionable and not solid enough to convince. Specifically, the comparison versus competing approaches is incomplete, both in terms of approaches considered and in the way these approaches have been tested. There is a single benchmark considered, and it has also been modified, making fair comparison with previous literature not feasible.